# A proline-rich motif on VGLUT1 reduces synaptic vesicle super-pool and spontaneous release frequency

Xiao Min Zhang[1,2,3‡], Urielle François[1,2†], Kätlin Silm[4†§], Maria Florencia Angelo[1,2], Maria Victoria Fernandez-Busch[1,2], Mona Maged[1,2], Christelle Martin[1,2], Véronique Bernard[4], Fabrice P Cordelières[5], Melissa Deshors[1,2], Stéphanie Pons[6], Uwe Maskos[6], Alexis Pierre Bemelmans[7], Sonja M Wojcik[3], Salah El Mestikawy[4,8], Yann Humeau[1,2], Etienne Herzog[1,2]*

[1]Interdisciplinary Institute for Neuroscience, Université de Bordeaux, Bordeaux, France; [2]Interdisciplinary Institute for Neuroscience CNRS UMR 5297, Bordeaux, France; [3]Department of Molecular Neurobiology, Max Planck Institute of Experimental Medicine, Göttingen, Germany; [4]Neuroscience Paris Seine NPS, Université Pierre et Marie Curie INSERM U1130 CNRS UMR8246, Paris, France; [5]Bordeaux Imaging Center, Université de Bordeaux, CNRS UMS 3420, INSERM US4, Bordeaux, France; [6]Institut Pasteur, CNRS UMR 3571, Unité NISC, Paris, France; [7]Commissariat à l'Energie Atomique et aux Energies Alternatives (CEA), Direction de laRecherche Fondamentale (DRF), Institut de Biologie François Jacob (IBFJ), MolecularImaging Research Center (MIRCen), Fontenay-aux-Roses, France; [8]Department of Psychiatry, Douglas Mental Health University Institute, McGill University, Montreal, Canada

*For correspondence:
etienne.herzog@u-bordeaux.fr

†These authors contributed equally to this work

Present address: ‡Department of Neurochemistry, Graduate School of Medicine, The University of Tokyo, Tokyo, Japan; §Department of Neurology and Department of Physiology, University of California, San Francisco, United States

Competing interests: The authors declare that no competing interests exist.

**Abstract** Glutamate secretion at excitatory synapses is tightly regulated to allow for the precise tuning of synaptic strength. Vesicular Glutamate Transporters (VGLUT) accumulate glutamate into synaptic vesicles (SV) and thereby regulate quantal size. Further, the number of release sites and the release probability of SVs maybe regulated by the organization of active-zone proteins and SV clusters. In the present work, we uncover a mechanism mediating an increased SV clustering through the interaction of VGLUT1 second proline-rich domain, endophilinA1 and intersectin1. This strengthening of SV clusters results in a combined reduction of axonal SV super-pool size and miniature excitatory events frequency. Our findings support a model in which clustered vesicles are held together through multiple weak interactions between Src homology three and proline-rich domains of synaptic proteins. In mammals, VGLUT1 gained a proline-rich sequence that recruits endophilinA1 and turns the transporter into a regulator of SV organization and spontaneous release.

DOI: https://doi.org/10.7554/eLife.50401.001

## Introduction

Synaptic vesicles (SVs) engage in multiple protein interactions at the presynaptic active zone (*Südhof and Rizo, 2011*) and fuse with the presynaptic plasma membrane upon calcium influx, to release their neurotransmitter content (*Lisman et al., 2007*). Within axon terminals, SVs are segregated from other organelles and grouped in a cluster behind the active zone (*Gray, 1959*). In adult neurons, SV supply at synapses depends not only on de novo vesicle biogenesis, but also on the exchange of mobile SVs between *en passant* boutons along the axon. This exchange pool has been

named 'SV super-pool' (*Kraszewski et al., 1996*; *Darcy et al., 2006*; *Westphal et al., 2008*; *Staras et al., 2010*; *Herzog et al., 2011*) and is probably a feature of both glutamatergic and GABAergic axons (*Wierenga et al., 2008*). While the last steps in the regulation of SV release have been studied intensively in different models, the relationship between super-pool SVs, clustered SVs, and the fine-tuning of release at terminals is much less well understood. However, synapsins, a family of SV associated phospho-proteins, play a central role in the regulation of SV clustering and mobility (*Pieribone et al., 1995*; *Song and Augustine, 2015*). A growing body of evidence furthermore suggests that SV cluster formation may result from a liquid phase separation from other cytoplasmic elements (*Milovanovic and De Camilli, 2017*; *Milovanovic et al., 2018*). Phase separation may be induced by the loose interaction of multiple proline-rich (or Poly-Proline; PRD) domains with multiple SH3 (Src Homology 3) domain proteins (*Li et al., 2012*). Indeed, PRD/SH3 interactions are numerous among the actors of SV trafficking such as synapsins and dephosphins (*Slepnev and De Camilli, 2000*; *Pechstein and Shupliakov, 2010*). In addition to these interactions, the actin cytoskeleton may contribute to the scaffolding of SV clusters and SV super-pool motility (*Darcy et al., 2006*; *Morales et al., 2000*; *Sankaranarayanan et al., 2003*; *Shupliakov et al., 2002*; *Gramlich and Klyachko, 2017*).

Besides SV dynamics and release competence, SV loading with neurotransmitter is another important parameter for the fine-tuning of neurotransmission. To fulfill this function, each excitatory SV may contain between 4 and 14 molecules of Vesicular Glutamate Transporters (*Takamori et al., 2006*; *Mutch et al., 2011*). Three isoforms of VGLUTs have been identified, and named VGLUT1-3 (*Takamori et al., 2000*; *Bellocchio et al., 2000*; *Herzog et al., 2001*; *Fremeau et al., 2001*; *Gras et al., 2002*; *Schäfer et al., 2002*). They share a nearly identical glutamate transport mechanism (*Schenck et al., 2009*; *Preobraschenski et al., 2014*; *Eriksen et al., 2016*) but have distinct expression patterns (*Fremeau et al., 2004a*). VGLUT1 is predominantly expressed in pathways of the olfactory bulb, neo-cortex, hippocampus and cerebellum and is associated with low release probability, while VGLUT2 is strongly expressed in sub-cortical pathways of the thalamus and brainstem, and is preferentially associated with high release probability projections (*Varoqui et al., 2002*; *Fremeau et al., 2004b*). This observation raised questions regarding a potential role of the VGLUT transporters in tuning SV release probability.

Hence, soon after their initial characterization, VGLUT1 and −2 were suspected to bear additional functional features that influence neurotransmitter release beyond quantal size (*Fremeau et al., 2004b*; *Wojcik et al., 2004*; *Moechars et al., 2006*; *Wallén-Mackenzie et al., 2006*). We discovered that mammalian VGLUT1, but not −2 or −3, interacts with the SH3 domain of endophilinA1 via a proline-rich sequence (*Vinatier et al., 2006*; *Voglmaier et al., 2006*; *De Gois et al., 2006*). The VGLUT1/EndophilinA1 interaction reduces SV release probability (*Weston et al., 2011*) and increases the speed of endocytosis of several SV proteins upon long trains of stimulation (*Voglmaier et al., 2006*; *Pan et al., 2015*). Furthermore, VGLUTs bear several di-leucine motifs on their N- and C- terminal sequences, which are responsible for efficient internalization after exocytosis (*Voglmaier et al., 2006*; *Pan et al., 2015*; *Foss et al., 2013*; *Li et al., 2017*). The functional relevance of these additional properties of VGLUTs is underscored by the 40% reduction in the number of SVs at *Slc17a7* (VGLUT1) knock-out (VGLUT1 KO) hippocampal Schaffer collateral and cerebellar parallel fiber terminals (*Fremeau et al., 2004b*; *Siksou et al., 2013*). Despite this reduction in clustered SV numbers, SV protein expression is not diminished nor displaced to other subcellular compartments (*Siksou et al., 2013*). Yet, we unexpectedly discovered a significantly larger SV super-pool in the axons of VGLUT1 KO neurons (*Siksou et al., 2013*). Also, VGLUT1 KO SVs appear pleomorphic under hyperosmotic chemical fixation (*Siksou et al., 2013*; *Herman et al., 2014*), however this latter phenotype is most likely a direct consequence of a major change in the ionic composition of the SV lumen (*Schenck et al., 2009*; *Preobraschenski et al., 2014*; *Eriksen et al., 2016*; *Martineau et al., 2017*).

In the present work, we determined the minimal domain responsible for the influence of VGLUT1 on the axonal super-pool in mammals. To this end, we generated mutants that disconnect the transport function of VGLUT1 from its trafficking function. We observed that VGLUT1 reduces SV super-pool size as well as the frequency of miniature Excitatory Post-Synaptic Currents (mEPSC) exclusively through the interaction of its PRD2 motif with EndophilinA1. These effects were further mediated by the non-canonical interaction of the VGLUT1/EndophilinA1 complex with the SH3B domain of the presynaptic scaffold protein intersectin1 (*Slepnev and De Camilli, 2000*). Taken together, our data

support the idea that VGLUT1 fine-tunes SV release by strengthening the liquid phase-separation between clustered and super-pool SVs.

## Results

### A specific and dose dependent reduction of super-pool size by VGLUT1

We first compared SV exchange between clusters and the axonal super-pool in wild type and VGLUT1 KO littermate primary neuron cultures. To this end, we expressed a tagged synaptobrevin protein (Syb2$^{EGFP}$) as a reporter (*Figure 1A*). Through FRAP (fluorescence recovery after photo-bleaching) of boutons we measured higher exchange rates of Syb2$^{EGFP}$ with the axonal compartment, for VGLUT1 KO neurons compared to wild-type neurons (*Figure 1B,C*, *Supplementary file 1*). A higher exchange rate of VGLUT1 KO SVs with the axonal super-pool is in line with our previous findings (*Siksou et al., 2013*). We then transduced VGLUT1$^{Venus}$ cDNA (*Herzog et al., 2011*) to rescue VGLUT1 KO neurons or over-express VGLUT1 in wild type neurons (*Figure 1D*). FRAP of VGLUT1$^{venus}$ fluorescence revealed that VGLUT1 overexpression further reduces SV exchange with axonal pools compared to the rescue of the knock-out to endogenous levels (*Figure 1E*). Finally, VGLUT2$^{venus}$ expression didn't reduce the VGLUT1 KO larger super-pool phenotype (*Figure 1—figure supplement 1*, *Supplementary file 1*). Therefore, VGLUT1 expression in neurons reduces the size of the SV super-pool in a dose dependent manner.

### Structure analysis of VGLUT1

To uncover the molecular mechanism by which VGLUT1 regulates SV super-pool size, we generated a series of mutants spanning the sequence of the transporter (*Figure 2A* and *Table 1*). VGLUT1 contains 12 trans-membrane domains with both termini on the cytoplasmic side and N-glycosylation on the first luminal loop (*Almqvist et al., 2007*). As vesicular glutamate transport strongly impacts SV tonicity, we first aimed at determining whether the SV loading state impacts SV mobility between SV clusters and the axonal super-pool. A triple point mutant R80Q, R176K, R314Q was generated to produce a glutamate transport deficient transporter as previously reported for VGLUT2 (*Herman et al., 2014*; *Almqvist et al., 2007*, sVGLUT1 for silent VGLUT1; *Figure 2A* blue residues; *Juge et al., 2006*).

We then focused our efforts on several conserved patterns at the VGLUT1 C-terminus. Indeed, mammalian VGLUT1 displays a unique double proline-rich (PRD1 530–540; PRD2 550–556) pattern conserved in all mammals and absent in VGLUT2 and −3 or in invertebrate orthologs of VGLUT1 (*Vinatier et al., 2006*). We thus generated deletions and point mutations to test the function of the PRD motifs (*Figure 2A* and *Table 1*). A conserved 540SYGAT sequence between PRD1 and PRD2 is present in all VGLUT isoforms including invertebrates (*Vinatier et al., 2006*). In addition to the deletion mutants, we also generated S540 and T544 to alanine mutations (*Figure 2A* and *Table 1*). We furthermore tested the full deletion of the C-terminus and point mutations in a putative PDZ-type3 binding domain (DQL514; PDZ: Post-synaptic density protein/*Drosophila* disc large tumor suppressor/Zonula occludens-1; see *Figure 2A* and *Table 1*).

All mutants were tagged using our successful c-terminal strategy (*Herzog et al., 2011*) and the expression level after transduction was monitored to match the endogenous levels of VGLUT1 using immunoblot (not shown). During FRAP and time-lapse experiments, the first frame of each sequence was used to further check the expression level of each mutant compared to the wild-type control (see examples in *Figure 2B* and quantification in *Figure 2C,D*). None of the mutants displayed a significant qualitative or quantitative difference in expression compared to wild-type controls.

### Vesicular glutamate uptake function does not influence SV super-pool size

VGLUT1 KO neurons display hypotonic SVs due to the loss of glutamate transport function (*Siksou et al., 2013*). To monitor a potential effect of SV lumen tonicity on SV mobility, we transduced VGLUT1$^{mCherry}$ or the triple mutation sVGLUT1$^{mCherry}$ together with Syb2$^{EGFP}$ in VGLUT1 KO neurons (*Figure 3A*) and probed SV turn over at synapses using Syb2$^{EGFP}$ FRAP. While we had a very high percentage of co-transduced neurons, we could find fibers with no VGLUT1$^{mCherry}$ signal and used these as a negative control. Both rescue conditions lowered the exchange of SVs

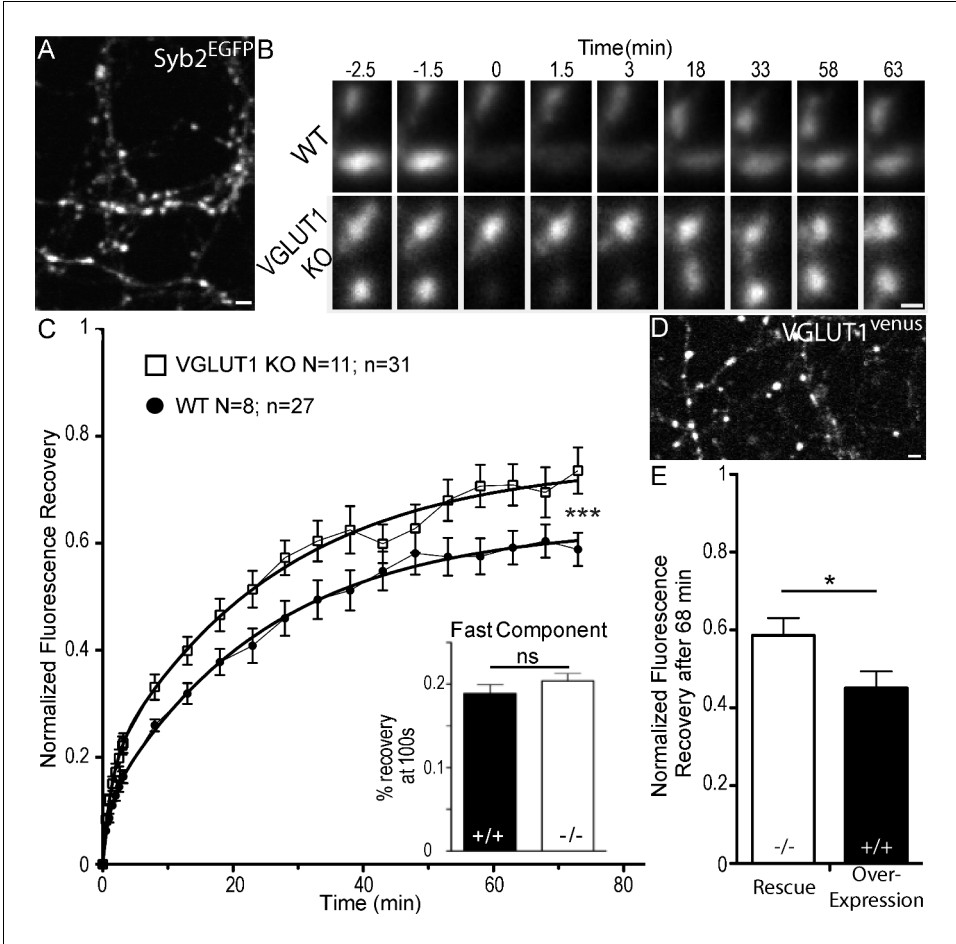

**Figure 1.** Dose dependent regulation of SV super-pool size by VGLUT1. (**A**) Expression of Synaptobrevin2 fused to Enhanced Green Fluorescent Protein (Syb2EGFP) in hippocampal neurons at 18 days in culture. (**B**) examples of FRAP sequences from both *Slc17a7* (VGLUT1) +/+ and - /- genotypes. Boutons are imaged for 3 min, bleached, and recovery is recorded for 73 min. (**C**) average FRAP kinetics. 27 synapses from +/+ and 31 synapses from - /- were measured by FRAP and the average traces are displayed here (for +/+ $N = 8$ cultures; for - /- $N = 11$ cultures). The two traces were fitted using double exponential components equations and the convergence of the traces to a common fit was tested using the extra sum of squares F test. The F test indicates that the traces are best fitted by two divergent models (*F* ratio = 19.32; p<0.0001). Fast FRAP recovery was monitored every 5 s in an independent set of experiments (inset). (**D**) Expression of transduced VGLUT1venus in hippocampal neurons. (**E**) Average FRAP recovery of VGLUT1venus at 68 min post-bleach in rescue or over-expression. Over-expression reduces the mobility of SVs (Unpaired *t* test, p=0.0385, t = 2.176. For overexpression: $N = 4$ cultures, $n = 14$ synapses; for rescue: $N = 3$ cultures, $n = 15$ synapses). scale bar: 2 μm in A and D, 1 μm in B.

DOI: https://doi.org/10.7554/eLife.50401.002

The following source data and figure supplements are available for figure 1:

**Source data 1.** Raw data for FRAP experiments with VGLUT1 WT and KO culture.

DOI: https://doi.org/10.7554/eLife.50401.004

**Source data 2.** Raw data for FRAP experiments with VGLUT1 rescued and VGLUT1 overexpressed culture.

DOI: https://doi.org/10.7554/eLife.50401.005

**Figure supplement 1.** VGLUT2 does not rescue SV super-pool size in VGLUT1 knock out neurons.

DOI: https://doi.org/10.7554/eLife.50401.003

**Figure supplement 1—source data 1.** Raw data for FRAP experiments with VGLUT1 and VGLUT2 rescued culture.

DOI: https://doi.org/10.7554/eLife.50401.006

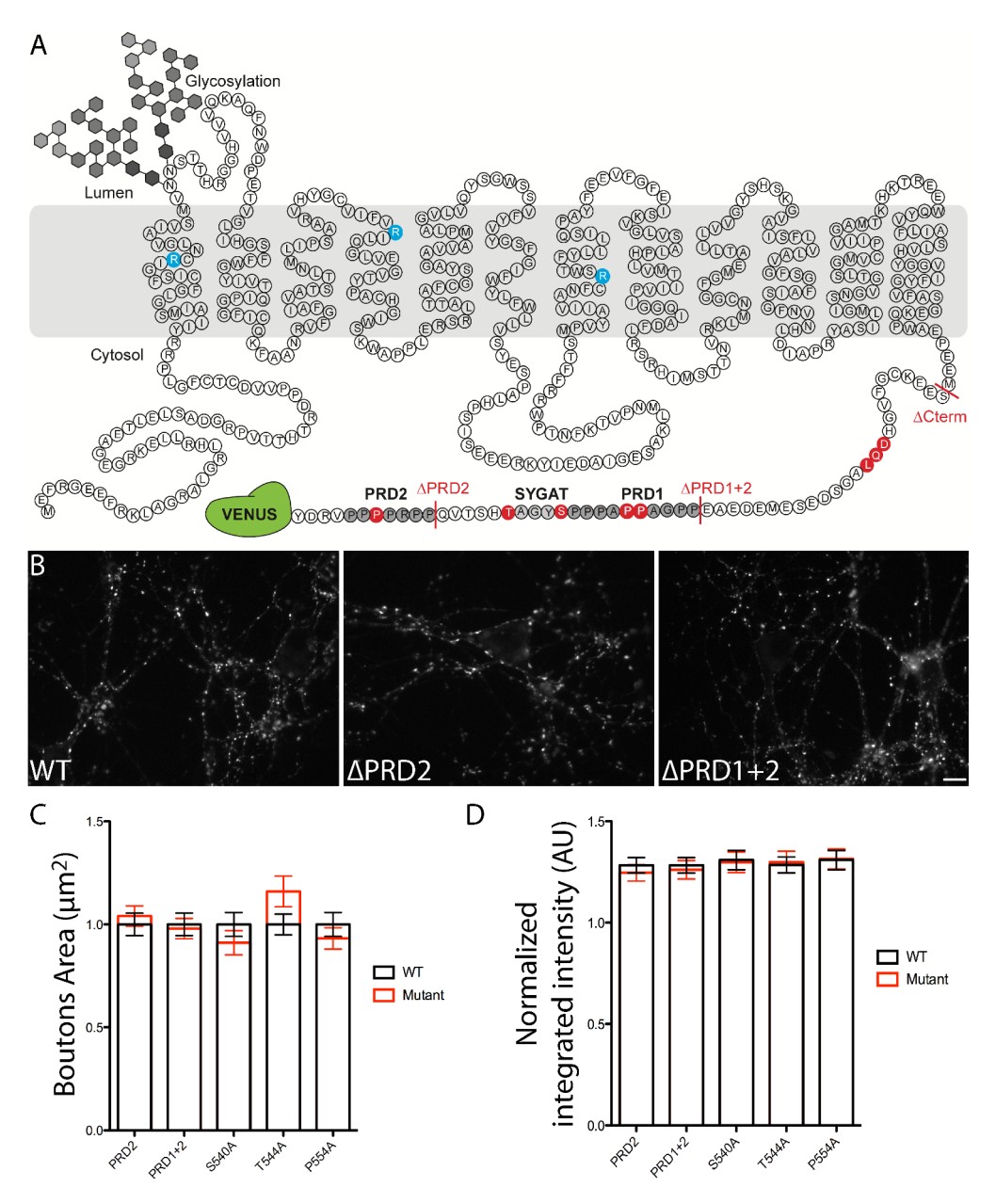

**Figure 2.** Structure of VGLUT1 and expression of mutants in neurons. (**A**) schematic of VGLUT1 structure with 560 amino acids, 12 transmembrane domains, N- and C-termini facing the cytoplasmic side, and N-glycosylation at the first luminal loop. In blue the three residues mutated to silence VGLUT1 transport (sVGLUT1 see *Figure 3*). Red bars mark the three deletions used (ΔC-term, ΔPRD1+2, ΔPRD2). Red residues were mutated to alanine in order to test their role in the super-pool regulation supported by VGLUT1. All mutations carried a venus tag at the C-terminus. (**B**) examples of expression patterns obtained with VGLUT1 WT and mutants upon transduction in hippocampal neurons matching endogenous expression levels. Note the dense punctate expression and low somatic signal typical of VGLUT1 distribution. Scale bar 10 μm. (**C**) Measurement of bouton area. None of the mutants displayed a shift in bouton size compared to WT controls. (**D**) Measurement of fluorescence intensity. None of the mutants displayed a significant shift in fluorescence intensity compared to WT controls.

DOI: https://doi.org/10.7554/eLife.50401.007

The following source data is available for figure 2:

**Source data 1.** Raw data for Bouton size and intensity in VGLUT1 WT and mutants rescued culture.

DOI: https://doi.org/10.7554/eLife.50401.008

**Table 1.** List of mutant constructs tested.

| | Mutation | Domain/Motif | Name | Putative function |
|---|---|---|---|---|
| VGLUT1 | R80Q/R176K/R314Q | TM1;TM4;TM7 | sVGLUT1 | Glutamate transport |
| | Δ504–560 | Whole C-terminus | ΔC-term | SV/VGLUT Trafficking |
| | DQL514AQA | PDZ type three binding | DQL514AQA | Unknown |
| | Δ530–560 | Proline Rich Domains 1+2 | ΔPRD1+2 | SH3 domain binding |
| | PP534AA | Proline Rich Domain1 | PP534AA | SH3 domain binding |
| | S540A | SYGAT | S540A | Unknown |
| | T544A | SYGAT | T544A | Unknown |
| | Δ550–560 | Proline Rich Domain 2 | ΔPRD2 | Endophilin binding |
| | P554A | Proline Rich Domain 2 | P554A | Endophilin binding |
| | EndoA1 290–352 | SH3 | SH3 | Binds PRD and ITSN1 SH3B |
| | EndoA1 290-352$^{E329K,S336K}$ | SH3 | SH3$^{E329K,S336K}$ | Binds PRD and ITSN1 SH3B |
| | ITSN1 903–971 | SH3B | SH3B | Endo SH3 binding through E329 and S336 |

DOI: https://doi.org/10.7554/eLife.50401.009

compared to VGLUT1 KO synapses of the same cultures significantly and to the same degree (*Figure 3B*, *Supplementary file 1*). In patch clamp experiments, a remaining spontaneous activity was found in the knock-out cultures that can be attributed to a minor but significant expression of VGLUT2 in hippocampal neurons (*Wojcik et al., 2004*; *Miyazaki et al., 2003*; *Herzog et al., 2006*). To minimize this contribution we monitored VGLUT2 levels in the culture at several ages and established that the lower plateau is reached between DIV17 and DIV22 when we performed our imaging and patch clamp experiments (see *Figure 3—figure supplement 1*). As anticipated, sVGLUT1 was unable to rescue the mEPSC amplitude to the same level as the wild type control (*Figure 3C,D*). The deficiency in the rescue of glutamate transport by sVGLUT1 is also apparent in the frequency of mEPSC events, as empty SVs also cycle (*Wojcik et al., 2004*; *Schuske and Jorgensen, 2004*). No impact of sVGLUT1 was seen on mIPSC features (*Figure 3—figure supplement 2*, *Supplementary file 1*). Hence, the triple mutation sVGLUT1 failed to rescue SV loading with glutamate, but reduced SV super-pool size to the same extent as the wild-type control.

## VGLUT1 PRD2 domain reduces SV super-pool size and the frequency of miniature EPSCs

We then tested a series of VGLUT1$^{venus}$ mutations spanning the C-terminal cytoplasmic tail of the transporter to rescue the VGLUT1 KO large SV super-pool phenotype (see *Figure 2* and *Table 1*). In the FRAP paradigm, only the mutations disrupting the PRD2 sequence were not able to reduce SV exchange down to WT levels (*Figure 4A,B*). A point mutation in PRD2 (P554A) that blocks the interaction with endophilin is sufficient to increase both the plateau of recovery and the rate of exchange of SVs (shorter slow half-life for P554A, *Figure 4A*, *Supplementary file 1*). Similar shifts were observed upon deletion of PRD2 (not shown). To further measure SV super-pool size, we performed time-lapse imaging at high sampling rates (five images per seconds) and tracked VGLUT1$^{venus}$ axonal transport between synaptic boutons (*Figure 4C–E*). This assay confirmed a significantly larger mobile axonal super-pool of SVs for VGLUT1$^{P554A-venus}$ but not for VGLUT1$^{S540A-venus}$ compared to WT rescue (*Figure 4D*). Yet, no difference in axonal transport speed could be measured between the three

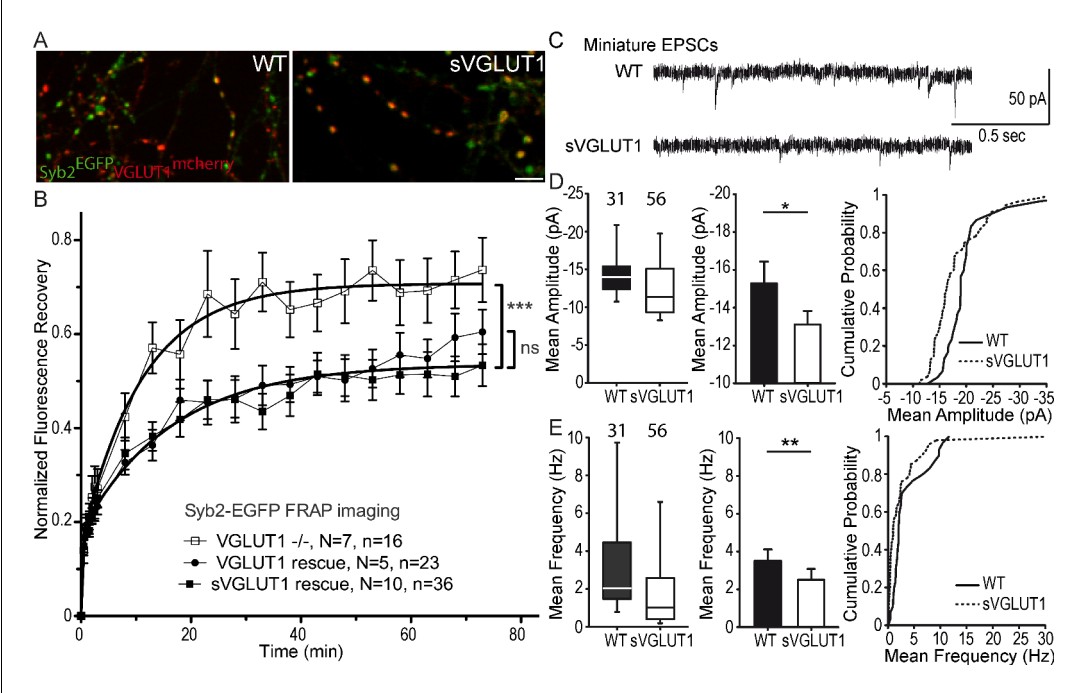

**Figure 3.** Glutamate transport and SV tonicity are not involved in the reduction of SV super-pool size. (**A**) Expression of VGLUT1$^{mCherry}$, sVGLUT1$^{mCherry}$ and Syb2$^{EGFP}$ in hippocampal neurons. FRAP was performed on Syb2$^{EGFP}$ at axons, both on those with, and those without mCherry signal. Scale bar 5 μm. (**B**) Average FRAP kinetics from knock-out cells rescued by VGLUT1$^{mCherry}$, sVGLUT1$^{mCherry}$, and VGLUT1 KO synapses not rescued. Synapses from each genotype were measured by FRAP and the average traces are displayed (N = 7 independent cultures, 16 synapses for -/-; N = 5 cultures, 23 synapses for WT rescue, and N = 10 cultures, 36 synapses for sVGLUT1 rescue). The three traces were fitted using double exponential components equations and the convergence of the traces to a common fit was tested using the extra sum of squares F test. The F test indicates that the traces are best fitted by two divergent models (one for - /- synapses and the other one for both rescues, F ratio = 30.25; p<0.0001). FRAP kinetics for the 2 types of rescued synapses are best fitted by one convergent model (F ratio = 1.235; p=0.294). (**C**) Spontaneous excitatory activity in VGLUT1 KO rescued neurons. Example traces of mEPSC activity in wild type and sVGLUT1 rescue conditions. (**D**) Comparison of the amplitude of mEPSC events in wild type and sVGLUT1 rescue conditions (N = 3 independent cultures, n = 31 cells, mean amplitude = 15.3 pA ± 1.15 SEM for wild type rescue; N = 3 independent cultures, n = 56 cells, mean amplitude = 13.12 pA ± 0.71 SEM for sVGLUT1 rescue; unpaired t-test p=0.011). (**E**) Comparison of the frequency of mEPSC events in wild type and sVGLUT1 rescue conditions. (N = 3 independent cultures, n = 31 cells, mean frequency = 3.51 Hz ± 0.6 SEM for wild type rescue; N = 3 independent cultures, n = 56 cells, mean frequency = 2.51 Hz ± 0.58 SEM for sVGLUT1 rescue; unpaired t-test p=0.008). Note that sVGLUT1 conditions are both significantly smaller than wild type rescue conditions.

DOI: https://doi.org/10.7554/eLife.50401.010

The following source data and figure supplements are available for figure 3:

**Source data 1.** Raw data for FRAP experiments with VGLUT1 WT, sVGLUT1 rescued and non-rescued culture.
DOI: https://doi.org/10.7554/eLife.50401.013

**Source data 2.** Raw data for Electrophysiological recording with VGLUT1 WT and sVGLUT1 rescued culture.
DOI: https://doi.org/10.7554/eLife.50401.014

**Figure supplement 1.** VGLUT2 expression diminishes in hippocampal neurons until DIV17.
DOI: https://doi.org/10.7554/eLife.50401.011

**Figure supplement 1—source data 1.** Raw data for Electrophysiological recording with VGLUT1 WT and sVGLUT1 rescued culture.
DOI: https://doi.org/10.7554/eLife.50401.015

**Figure supplement 2.** Vesicular glutamate uptake function does not influence mIPSCs features.
DOI: https://doi.org/10.7554/eLife.50401.012

constructs (*Figure 4E*). We also investigated whether PRD2 function could be regulated by phosphorylation of the conserved 540-SYGAT sequence. We thus implemented the PhosTag assay that specifically shifts the electrophoretic mobility of phospho-proteins (*Kinoshita et al., 2006*). VGLUT1-$^{venus}$ constructs displayed an additional slow band in PhosTag gel migration, except when samples were digested with alkaline phosphatase (*Figure 4—figure supplement 1C*). All mutants including ΔC-term displayed this second slower band. Thus, VGLUT1 appears to be phosphorylated, but not

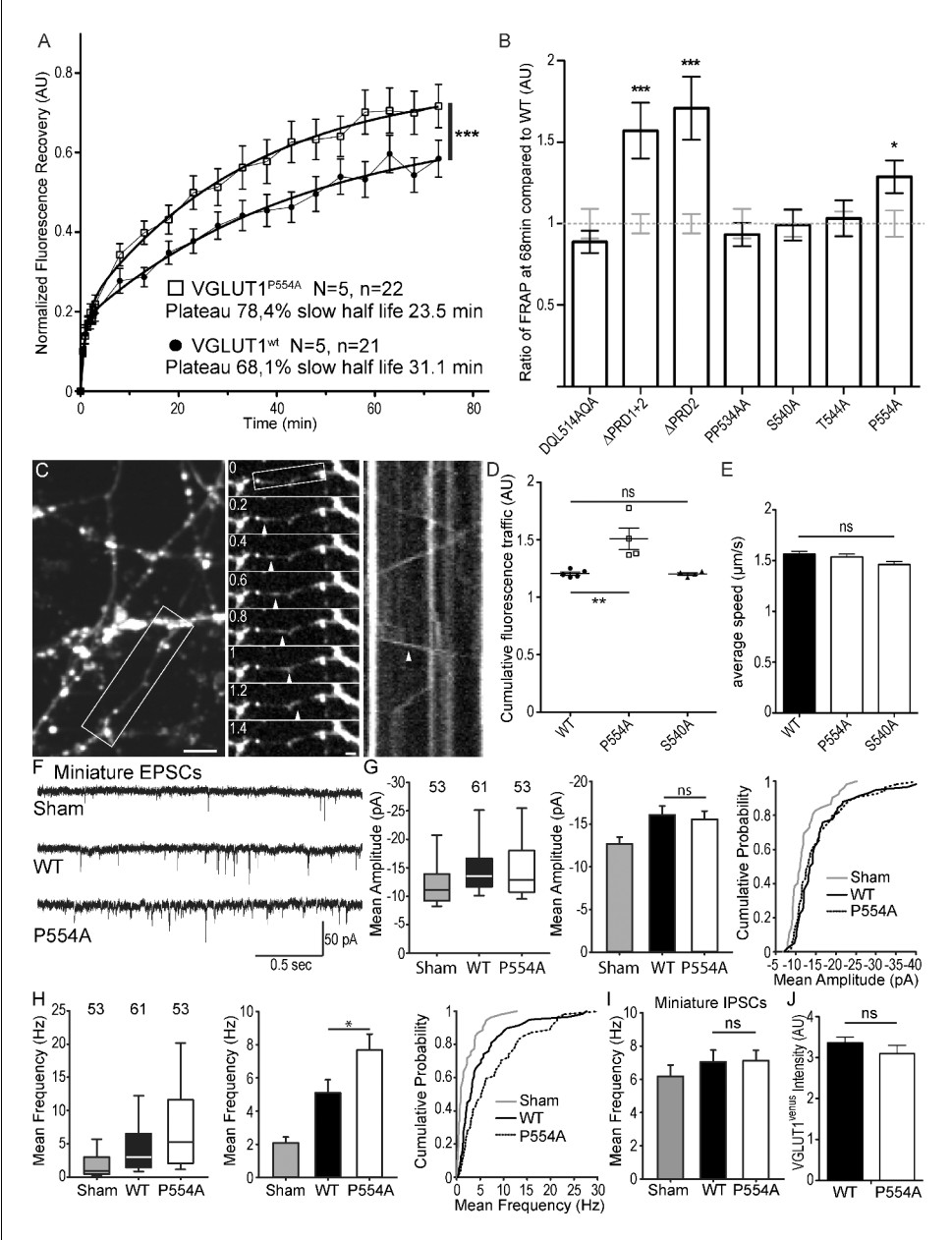

**Figure 4.** The VGLUT1 PRD2 domain mediates SV super-pool size and mEPSC frequency reductions. (**A**) Comparison of VGLUT1[venus] and VGLUT1[P554A-venus] rescues of VGLUT1 KO on SV exchange rates at synapses. 21 (WT) and 22 (P554A) synapses from each rescue were measured by FRAP and the average traces are displayed (*N* = 5 cultures for WT and *N* = 5 cultures for P554A). The two traces were fitted using double exponential components equations and the convergence of the traces to a common fit was tested using the extra sum of squares F test. The F test indicates that the traces are best fitted by two divergent models (*F* ratio = 19.64; p<0.0001). (**B**) Similar experiments where performed for ΔPRD1+2, ΔPRD2, DQL514AQA, PP534AA, S540A and T544A. The results are displayed here as a comparison of fluorescence recovery to the corresponding WT control 68 min after bleaching. Only mutants affecting PRD2 display a lack of reduction to WT levels and a significantly higher SV exchange rate (*t* test, WT *vs.* ΔPRD1+2: p=0.0005, *t* = 3.790; *vs.* ΔPRD2: p<0.0001, *t* = 4.369; *vs.* P554A: p=0.0293, *t* = 2.266). (**C**) Time lapse imaging of SV axonal transport. VGLUT1[venus], or VGLUT1[P554A-venus], or VGLUT1[S540A-venus] were expressed in VGLUT1 KO neurons. Example sequence extracted from the boxed fiber (left) sampled every 200 ms over 30 s (middle). Vertical arrowhead points to a venus fluorescent dot traveling along the axon. Kymograph of fluorescence movements within the example fiber (boxed in middle panel). Vertical arrowhead points to the same traced event shown in the middle panel. Scale bar: left 5 μm, middle 2 μm. (**D**) Cumulative axonal fluorescence traffic measured over time-lapse sequences. A significant increase in VGLUT1[venus] traffic is seen when P554A is expressed compared to WT and S540A mutant (unpaired *t* test, WT *vs.* P554A: p=0.0092, *t* = 3.564; *vs.* S540A: p=0.8963, *t* = 0.1351. *N* = 5 cultures for WT, and *N* = 4 cultures for both P554A and S540A). (**E**) Average speed of VGLUT1[venus] dots was extracted from kymographs. No significant changes in speed was seen for mutants tested compared to WT (One-way ANOVA, p=0.0634, *F* ratio = 2.431. Speed for each mutant, WT: 1.564 ± 0.02692 μm/s; P554A: 1.536 ± 0.02860 μm/s; S540A: 1.460 ± 0/02942 μm/s). (**F**) Spontaneous excitatory activity in VGLUT1 KO

*Figure 4 continued on next page*

*Figure 4 continued*

neurons rescued by wild type and P554A VGLUT1 constructs. Example traces of mEPSC activity in sham controls, wild type and VGLUT1[P554A] rescue conditions. (G) Comparison of the amplitude of mEPSC events in wild type and VGLUT1[P554A] rescue conditions (*N* = 3 independent cultures, n = 53 cells, mean amplitude = 12.73 pA ± 0.79 SEM for sham controls; *N* = 3 independent cultures, n = 61 cells, mean amplitude = 16.05 pA ± 1.05 SEM for wild type rescue; *N* = 3 independent cultures, n = 53 cells, mean amplitude = 15.59 pA ± 1.02 SEM for *vglut1[P554A]* rescue; Mann-Whitney test of wild type versus VGLUT1[P554A], p=*0.485*). Note that both wild type and VGLUT1[P554A] rescue mEPSC amplitudes significantly and to the same extent compared to sham controls. (H) Comparison of the frequency of mEPSC events in wild type and VGLUT1[P554A] rescue conditions (*N* = 3 independent cultures, n = 53 cells, mean frequency = 2.14 Hz ± 0.37 SEM for sham controls; *N* = 3 independent cultures, n = 61 cells, mean frequency = 5.53 Hz ± 0.87 SEM for wild type rescue; *N* = 3 independent cultures, n = 53 cells, mean frequency = 7.59 Hz ± 0.97 SEM for VGLUT1[P554A] rescue; Mann-Whitney test of wild type versus VGLUT1[P554A], p=*0.031*). Note that VGLUT1[P554A] mEPSC events are significantly more frequent than wild type rescue conditions. (I) Comparison of the frequency of mIPSC events in wild type and VGLUT1[P554A] rescue conditions (*N* = 3 independent cultures, n = 53 cells, mean frequency = 6.21 Hz ± 0.69 SEM for sham controls; *N* = 3 independent cultures, n = 61 cells, mean frequency = 7.28 Hz ± 0.69 SEM for wild type rescue; *N* = 3 independent cultures, n = 53 cells, mean frequency = 6.79 Hz ± 0.63 SEM for VGLUT1[P554A] rescue; Mann-Whitney test of wild type versus VGLUT1[P554A], p=*0.781*). Note that all groups are equivalent regarding mIPSC amplitudes (*Figure 4—figure supplement 2* displays the full IPSC data set). (J) Post-hoc analysis of VGLUT1[venus] average fluorescence integrated intensity at boutons in cultures monitored in electrophysiology. (*N* = 2 independent cultures, mean integrated intensity of punctate VGLUT1[venus]signal = 3.28 AU ± 0.15 SEM for wild type rescue; *N* = 2 independent cultures, mean integrated intensity of punctate VGLUT1[venus]signal = 3.11 AU ± 0.31 SEM for VGLUT1[P554A] rescue; *T* test, p=*0.348*).

DOI: https://doi.org/10.7554/eLife.50401.016

The following source data and figure supplements are available for figure 4:

**Source data 1.** Raw data for FRAP experiments with VGLUT1 WT and P554A rescued culture.
DOI: https://doi.org/10.7554/eLife.50401.019
**Source data 2.** Raw data for FRAP experiments with VGLUT1 WT, ΔPRD 1+2 and ΔPRD 2 rescued culture.
DOI: https://doi.org/10.7554/eLife.50401.020
**Source data 3.** Raw data for FRAP experiments with VGLUT1 WT and DQL514AQA rescued culture.
DOI: https://doi.org/10.7554/eLife.50401.021
**Source data 4.** Raw data for FRAP experiments with VGLUT1 WT and PP534AA rescued culture.
DOI: https://doi.org/10.7554/eLife.50401.022
**Source data 5.** Raw data for FRAP experiments with VGLUT1 WT and S540A rescued culture.
DOI: https://doi.org/10.7554/eLife.50401.023
**Source data 6.** Raw data for FRAP experiments with VGLUT1 WT and T544A rescued culture.
DOI: https://doi.org/10.7554/eLife.50401.024
**Source data 7.** Raw data for Cumulative SV axonal transport in VGLUT1 WT, P554A and S540A rescued culture.
DOI: https://doi.org/10.7554/eLife.50401.025
**Source data 8.** Raw data for SV axonal transport speed in VGLUT1 WT, P554A and S540A rescued culture.
DOI: https://doi.org/10.7554/eLife.50401.026
**Source data 9.** Raw data for Electrophysiological recording with VGLUT1 WT and P554A rescued and non-rescued culture.
DOI: https://doi.org/10.7554/eLife.50401.027
**Figure supplement 1.** VGLUT1 is phosphorylated but not at the conserved 540-SYGAT motif.
DOI: https://doi.org/10.7554/eLife.50401.017
**Figure supplement 2.** VGLUT1 PRD2 domain removal does not affect mIPSCs features.
DOI: https://doi.org/10.7554/eLife.50401.018
**Figure supplement 2—source data 1.** Raw data for Electrophysiological recording with VGLUT1 WT and P554A rescued and non-rescued culture.
DOI: https://doi.org/10.7554/eLife.50401.028

at the C-terminal tail, and the function of the extremely conserved 540-SYGAT sequence therefore remains unclear.

In patch-clamp experiments, we tested the effect of VGLUT1[P554A-venus] rescue on miniature events, compared to WT and empty vector controls (sham; *Figure 4F*). VGLUT1[P554A] rescued miniature EPSC amplitudes to a similar level as the wild type transporter (*Figure 4G*). Interestingly, VGLUT1[P554A] increased the frequency of miniature events significantly more than the WT rescue (*Figure 4H*). In contrast to this, mIPSC frequencies were similar in all three conditions (*Figure 4I* and *Figure 4—figure supplement 2*, *Supplementary file 1*). As intended in our rescue strategy, VENUS fluorescence intensity at boutons was not significantly different between groups tested by electrophysiology (*Figure 4J*). Therefore, the PRD2 sequence of mammalian VGLUT1 is sufficient to reduce SV super-pool size and mEPSC frequency.

# Intersectin1 interaction with endophilin A1 mediates the reduction of the SV super-pool promoted by VGLUT1 expression

Finally, we performed competition experiments with 3 SH3 domains to investigate the VGLUT1[PRD2] dependent pathway mediating SV super pool reduction (see *Table 1* and *Figure 5A*). SH3 domains fused to the cyan fluorescent protein mCERULEAN3 (*Markwardt et al., 2011*) were overexpressed under control of the human synapsin promoter in neuron cultures from VGLUT1[venus] mice (*Figure 5B*). Single boutons from fibers coexpressing mCERULEAN3 and VENUS were selected for FRAP experiments. An empty mCERULEAN3 sham construct was used as negative control and had a VGLUT1[venus] FRAP curve similar to our previous measurements (*Figure 5B,C*). The SH3 domain of EndophilinA1 was used to displace endogenous full-length EndophilinA1 to test the contribution of the membrane binding Bin1, Amphiphysin, RVs (BAR) domain. The FRAP curve in this experiment matched the sham control recovery (*Figure 5B,C*). The SH3 domain of EndophilinA1 mutated at E329K and S336K was over-expressed to displace endogenous EndophilinA1 and block the interaction with the SH3B domain of intersectin1 (*Pechstein et al., 2015*). Endo-SH3[E329K,S336K] shifted the

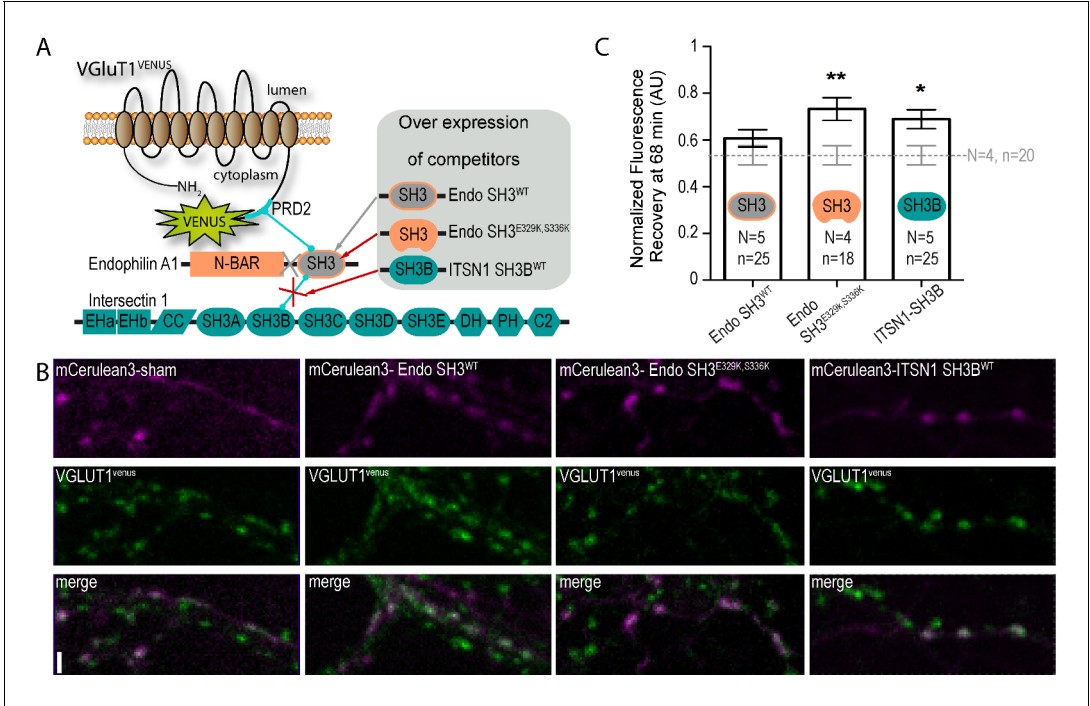

**Figure 5.** A tripartite complex between VGLUT1, endophilinA1 and intersectin1 mediates SV super-pool reduction. (**A**) Schematic model of the competition experiment designed to test for the effective recruitment of intersectin1 at the VGLUT1/endophilinA1 complex (*Pechstein et al., 2015*). The three SH3 domains over-expressed in this competition assay disrupt distinct parts of the tripartite complex. The SH3 domain of endophilinA1 displaces the endogenous endophilinA1 BAR domain (Endo SH3[WT]). The SH3 domain of endophilinA1 mutated at E329K and S336K disrupts the interaction of intersectin1 with the VGLUT1/endophilinA1 complex (*Pechstein et al., 2015*). Finally, the SH3B domain of intersectin1 should displace the endogenous intersectin1 and allows us to assess whether the full-length intersectin1 is required. (**B**) Over-expression of SH3 domains fused with mCerulean3 in VGLUT1[venus] neurons. Axons filled with mCerulean3 were selected for FRAP experiments. Scale bar 2 μm. (**C**) FRAP measurement of SV super-pool in SH3 domains competition experiments. Synapses from each over-expression condition were measured by FRAP and the average recovery after 68 min are displayed ($0.5346 \pm 0.04128$, $N = 4$ and $n = 20$ for sham; $0.6071 \pm 0.03705$, $N = 5$ and $n = 25$ for Endo-SH3[WT]; $0.7323 \pm 0.04909$, $N = 4$ and $n = 18$ for Endo-SH3[E329K,S336K]; $0.6894 \pm 0.04036$, $N = 5$ and $n = 25$ for ITSN1-SH3B). Sham and Endo-SH3[WT] converged to a lower SV exchange recovery (Mann-Whitney test, $p=0.141$) while Endo-SH3[E329K,S336K] (Mann-Whitney test, $p=0.005$ **) and ITSN1-SH3B (Mann-Whitney test, $p=0.014$ *) generated higher SV exchange recovery curves.

DOI: https://doi.org/10.7554/eLife.50401.029

The following source data and figure supplement are available for figure 5:

**Source data 1.** Raw data for FRAP experiments with SH3 domain mutant overexpressed VGLUT1venusculture.

DOI: https://doi.org/10.7554/eLife.50401.031

**Figure supplement 1.** EndophilinA1 accumulates at synaptic vesicle clusters of VGLUT1 synapses.

DOI: https://doi.org/10.7554/eLife.50401.030

FRAP recovery to a significantly higher plateau, creating a phenocopy of the VGLUT1[P554A] and VGLUT1 KO mutants (*Figure 5B,C*, *Supplementary file 1*). Similarly, the SH3B domain of intersectin1 was over-expressed to displace endogenous intersectin1 from EndophilinA1. ITSN1-SH3B also shifted the FRAP recovery to a higher plateau. Hence, full length intersectin1 may be required for the VGLUT1 mediated reduction of SV super pool (*Figure 5B,C*). Taken together, these competition experiments reveal the involvement of EndophilinA1 and interectin1 in a complex with VGLUT1 in the negative regulation of SV super pool size in mammals.

## Discussion

In the present study we could observe that VGLUT1 PRD2 motif is a negative regulator of SV mobility and mEPSC frequency. Our data support that SV super-pool reduction is mediated by an interaction of endophilinA1 with both VGLUT1 PRD2 and intersectin1 SH3B domains.

### Molecular dissection of VGLUT1 functions

Previous works had shown that several dileucine-like motifs are involved in VGLUT1 endocytosis (*Voglmaier et al., 2006*; *Pan et al., 2015*; *Foss et al., 2013*) and that the PRD2 of VGLUT1 regulates endocytosis during long trains of stimulation (*Voglmaier et al., 2006*). The present data provide the first molecular separation of glutamate transport and SV trafficking functions of VGLUT1. The sVGLUT1 mutant allowed us to test the function of the VGLUT1 backbone structure without potential interference brought by the complex flow of ions occurring through glutamate loading (*Schenck et al., 2009*; *Preobraschenski et al., 2014*; *Eriksen et al., 2016*; *Juge et al., 2006*; *Goh et al., 2011*). sVGLUT1 revealed that glutamate transport is not related to the negative regulation of SV super-pool size. Instead, our detailed structure-function analysis of VGLUT1 C-terminus showed that the PRD2 motif is necessary and sufficient to inhibit SV exchange between boutons and the axonal super-pool. We further tested a conserved putative type 3 PDZ binding motif (DQL514) and the PRD1 pattern but found no significant effect on SV super-pool size. Though we brought evidence for VGLUT1 phosphorylation, putative phosphorylation is not at the C-terminus. It thus remains to be seen whether additional features of the C-terminus such as 540SYGAT can be shown to regulate VGLUT1 tuning of SV mobility.

Hence, distinctively from the dileucine based endocytosis of VGLUTs, the C-terminal proline-rich domain PRD2 supports the negative regulation of SV mobility and mEPSC frequency in mammals. SV mobility inhibition is operated independently of glutamate loading processed by the core transmembrane domains. The structure-function relationship of other conserved sequences remain to be established.

### Mammalian VGLUT1 is a molecular player of SV clustering at synapses

Synaptic vesicles are segregated from other organelles in the nerve terminal, and grouped in a cluster (*Gray, 1959*). It has been proposed that SVs are recruited to the cluster by synapsins to an actin based cytoskeleton (*Hilfiker et al., 1999*). Yet, several lines of evidence suggest a different mechanism involving the low affinity binding of many partners forming a liquid phase separation to the rest of the cytosol (*Milovanovic and De Camilli, 2017*; *Pechstein and Shupliakov, 2010*; *Sankaranarayanan et al., 2003*). Indeed, endophilinA1, intersectin, amphiphysin are abundant within the vesicle cluster (*Figure 5—figure supplement 1*; *Pechstein and Shupliakov, 2010*; *Evergren et al., 2007*) while the binding of SH3 domains to proline-rich motifs has been shown to generate liquid phase separations in a synthetic assay (*Li et al., 2012*). Further, synapsins were recently shown to induce liquid phase separation with lipid vesicles (*Milovanovic et al., 2018*). Hence, the best model to date infers the liquid phase separation of SVs in a dynamic array of labile interactions between synapsins, dephosphins (endophilins, amphiphysins, EPS15, synaptojanins...) and intersectins.

Previously, a slower re-acidification of PRD2 deleted VGLUT1-phluorin probes during long trains of stimulation suggested that the recruitment of endophilinA1 increases the endocytosis efficiency of VGLUT1 SVs (*Voglmaier et al., 2006*). However, recent reports indicate that endophilinA1 interaction with intersectin1 favors clathrin uncoating (*Pechstein et al., 2015*; *Milosevic et al., 2011*), whereas the clathrin coat was shown to inhibit SV acidification (*Farsi et al., 2018*). Therefore, the recruitment of endophilinA1 and intersectin1 at SVs may speed up clathrin uncoating and SV

acidification. This advocates for new experiments designed to discriminate between an impact of VGLUT1 PRD2 on SV endocytosis versus SV uncoating, consecutive acidification kinetics and clustering. More recently, intersectin1 (ITSN1) SH3B was shown to interact with the SH3 domain of endophilinA1 while preserving the interaction of endophilinA1 with its PRD targets on Dynamin and VGLUT1 (*Pechstein et al., 2015*). Furthermore, intersectin1 may as well interact directly with VGLUT1 through the SH3A domain (*Santos et al., 2014*; *Richter et al., 2018*). Our present data provide evidence for a reduction of the SV super-pool size to the benefit of the SV clusters mediated by a tripartite complex between VGLUT1 PRD2, endophilinA1 and intersectin1 (VGLUT1/EndoA1/ITSN1, *Figure 5*). The increase in SV exchange rate when PRD2 is mutated strongly suggests a reduced strength in the interactions scavenging SVs in the cluster (*Staras et al., 2010*; *Herzog et al., 2011*). Yet further investigations using single particle tracking methods will be necessary to address whether VGLUT1 PRD2 reduces SV mobility within synaptic clusters as well (*Gramlich and Klyachko, 2017*). Our data also fit well with recent reports of intersectin1 function in the re-clustering of newly endocytosed SVs, and on the nano-scale organization of synapsins at terminals (*Gerth et al., 2017*; *Winther et al., 2015*). Downstream of VGLUT1/EndoA1/ITSN1, the SV super-pool reduction may be driven by interactions of intersectin1 with synapsins (*Milovanovic et al., 2018*; *Winther et al., 2015*) and/or through remodeling of the actin cytoskeleton surrounding the cluster (*Humphries et al., 2014*). Yet further analysis will be required to discriminate between these pathways and evaluate a possible regulation by phosphorylation/dephosphorylation cycles.

## Mammalian VGLUT1 acts as a dual regulator of glutamate release

Vesicular glutamate transporters are necessary and sufficient to generate a glutamatergic phenotype in neurons by loading secretory organelles with glutamate (*Takamori et al., 2000*). Quantal size modulation has been reported upon changes in the level of VGLUT expression (*Wojcik et al., 2004*; *Moechars et al., 2006*; *Wilson et al., 2005*) and is reproduced in our present work (*Figures 3* and *4*). Furthermore, EndophilinA1 binding to the VGLUT1 PRD2 sequence was shown to reduce SV release probability (*Weston et al., 2011*). Our dataset now supports the role of VGLUT1/EndoA1/ITSN1 in the reduction of mEPSC frequency and SV exchange between clusters and super-pool (*Figures 4* and *5*). A discrepancy may arise from the fact that Weston et al. did not see an effect of VGLUT1 ΔPRD2 on mEPSC frequency but only on SV release probability. However, this may be explained by two differences in our recording conditions. First, the autaptic culture system from Weston et al. prevents the formation of a network. Autapses are powerful tools to dissect evoked activity, but mEPSCs in autaptic conditions arise from a single cell (the recorded cell), while in continental cultures they are generated by multiple cells targeting the recorded neuron. Additionally, continental cultures build a network that generates activity, which most likely leads to different setpoints of SV super-pool sizes and homeostatic plasticity compared to autaptic micro-islands (*De Gois et al., 2005*). Second, Weston and colleagues worked before 14 days in culture, whereas we worked after 17 days in culture when VGLUT2 expression reached a lower plateau and may generate less background mEPSC activity (*Figure 3—figure supplement 1*; *Wojcik et al., 2004*; *Herzog et al., 2006*).

The molecular mechanism of SV super-pool reduction that we uncovered requires the EndoA1 SH3 domain, but not the BAR domain (*Figure 5*). Previously, *Weston et al. (2011)* showed that endophilinA1 promotes SV release probability through dimerization and BAR domain mediated membrane binding. Our current results complement this model by adding a pathway through which VGLUT1/EndoA1 recruit intersectin1 to promote SV clustering and mEPSC frequency reduction in a BAR domain independent fashion. Hence, free EndophilinA1 may actively promote SV exocytosis through membrane binding mechanisms while VGLUT1-bound EndophilinA1 may actively reduce SV mobility and exocytosis through an intersectin1 dependent pathway.

Beyond setting the quantal size, VGLUT1 influences glutamate release parameters most likely by changing the strength of SVs phase separation in clusters. It remains to establish how neurons take advantage of this feature to locally modulate the parameters of quantal release at selected synapses (*Staras et al., 2010*; *Herzog et al., 2011*; *Rothman et al., 2016*). In this line, neuromodulation has been very recently proposed to tune SV pools in the axon through GPCR signaling and synapsin phosphorylation (*Patzke et al., 2019*).

# Materials and methods

## Key resources table

| Reagent type (species) or resource | Designation | Source or reference | Identifiers | Additional information |
|---|---|---|---|---|
| Strain, strain background (*Mus musculus*) | *Slc17a7*[-/-] (VGLUT1 KO) mice | PMID: 15103023 | available upon request to Dr Sonja M. Wojcik wojcik@em.mpg.de | |
| Strain, strain background (*Mus musculus*) | *Slc17a7*[v/v] (VGLUT1venus KI) mice | PMID: 22031900 | available upon request to corresponding author | |
| Transfected construct (*Homo sapiens*) | F(syn)W-RBN::Synaptobervin2-EGFP | PMID: 23581566 | available upon request to Dr. Etienne Herzog | *Figure 1*, *Figure 3* and related results part. Lentiviral vector expressing Syb2EGPF. |
| Transfected construct (*Rattus norvegicus*) | F(syn)W-RBN::VGLUT1-venus | PMID: 23581566 | available upon request to Dr. Etienne Herzog | *Figure 1*, *Figure 4* and related results part. Lentiviral vector expressing VGLUT1venus. |
| Transfected construct (*Rattus norvegicus*) | F(syn)W-RBN::VGLUT2-venus | This paper | available upon request to Dr. Etienne Herzog | *Figure 1—figure supplement 1–S1* and related results part. Lentiviral vector expressing VGLUT2venus. |
| Transfected construct (*Rattus norvegicus*) | F(syn)W-RBN::VGLUT1 ΔC-term-venus | This paper | available upon request to Dr. Etienne Herzog | *Figure 4* and related results part. Lentiviral vector expressing VGLUT1ΔC-term-venus. |
| Transfected construct (*Rattus norvegicus*) | F(syn)W-RBN::VGLUT1 DQL514AQA-venus | This paper | available upon request to Dr. Etienne Herzog | *Figure 4* and related results part. Lentiviral vector expressing VGLUT1DQL514AQA-venus. |
| Transfected construct (*Rattus norvegicus*) | F(syn)W-RBN::VGLUT1 ΔPRD1+2-venus | This paper | available upon request to Dr. Etienne Herzog | *Figure 4* and related results part. Lentiviral vector expressing VGLUT1ΔPRD1+2-venus. |
| Transfected construct (*Rattus norvegicus*) | F(syn)W-RBN::VGLUT1 PP534AA-venus | This paper | available upon request to Dr. Etienne Herzog | *Figure 4* and related results part. Lentiviral vector expressing VGLUT1PP534AA-venus. |

*Continued on next page*

*Continued*

| Reagent type (species) or resource | Designation | Source or reference | Identifiers | Additional information |
|---|---|---|---|---|
| Transfected construct (*Rattus norvegicus*) | F(syn)W-RBN:: VGLUT1 S540A-venus | This paper | available upon request to Dr. Etienne Herzog | *Figure 4* and related results part. Lentiviral vector expressing VGLUT1S540A-venus. |
| Transfected construct (*Rattus norvegicus*) | F(syn)W-RBN:: VGLUT1 T544A-venus | This paper | available upon request to Dr. Etienne Herzog | *Figure 4* and related results part. Lentiviral vector expressing  VGLUT1T544A-venus. |
| Transfected construct (*Rattus norvegicus*) | F(syn)W-RBN:: VGLUT1 ΔPRD2-venus | This paper | available upon request to Dr. Etienne Herzog | *Figure 4* and related results part. Lentiviral vector expressing VGLUT1ΔPRD2-venus. |
| Transfected construct (*Rattus norvegicus*) | F(syn)W-RBN:: VGLUT1 P554A-venus | This paper | available upon request to Dr. Etienne Herzog | *Figure 4* and related results part. Lentiviral vector expressing VGLUT1P554A-venus. |
| Transfected construct (*Rattus norvegicus*) | AAV9::VGLUT1mCherry-miniSOG | This paper | available upon request to Dr. Etienne Herzog | *Figure 3* and related results part. Lentiviral vector expressing VGLUT1mCherry-miniSOG. |
| Transfected construct (*Rattus norvegicus*) | AAV9:: sVGLUT1mCherry-miniSOG | This paper | available upon request to Dr. Etienne Herzog | *Figure 3* and related results part. Lentiviral vector expressing sVGLUT1mCherry-miniSOG. |
| Transfected construct (*Homo sapiens*) | AAV9::mCerulean3-EndophilinA1SH3 | This paper | available upon request to Dr. Etienne Herzog | *Figure 5* and related results part. Lentiviral vector expressing mCerulean3-EndophilinA1SH3. |
| Transfected construct (*Homo sapiens*) | AAV9:: mCerulean3-EndophilinA1 SH3$^{E329K, S336K}$ | This paper | available upon request to Dr. Etienne Herzog | *Figure 5* and related results part. Lentiviral vector expressing mCerulean3-EndophilinA1 SH3$^{E329K, S336K}$. |
| Transfected construct (*Homo sapiens*) | AAV9:: mCerulean3-IntersectinSH3B | This paper | available upon request to Dr. Etienne Herzog | *Figure 5* and related results part. Lentiviral vector expressing mCerulean3-IntersectinSH3B. |
| Antibody | GFP, Mouse, monoclonal | Roche | Cat. 11814460001 RRID:AB_390913 | 1:1000 |

*Continued on next page*

*Continued*

| Reagent type (species) or resource | Designation | Source or reference | Identifiers | Additional information |
|---|---|---|---|---|
| Antibody | VIAAT, Guinea pig, polyclonal | SYSY | Cat. 131004 RRID:AB_887873 | 1:1000 |
| Antibody | VGLUT2, Guinea pig, polyclonal | Millipore | Cat. AB2251 RRID:AB_1587626 | 1:2000 |
| Antibody | VGLUT1, guinea pig, polyclonal | Merck | Cat. AB5905 RRID:AB_2301751 | 1:5000 |
| Antibody | EndophilinA1, rabbit, polyclonal | PMID: 16606361 | available upon request to Dr. Etienne Herzog | 1:500 |
| Peptide, recombinant protein | FastAP Thermosensitive Alkaline Phosphatase | Thermo Scientific | Cat. EF0651 | |
| Peptide, recombinant protein | Halt phosphatase inhibitor cocktail | Thermo Scientific | Cat. 78420 | |
| Chemical compound, drug | Phos-tag Acrylamid | Wako | Cat. AAL-107 | |
| Software, algorithm | FRAP Analysis Plugin | This paper | | The plugin is available at: https://github.com/fabricecordelieres/IJ-Macro_FRAP-MM |
| Software, algorithm | KymoToolbox Plugin | PMID: 23374344 | | The plugin is available at: https://github.com/fabricecordelieres/IJ-Plugin_KymoToolBox |

## Animals

All *Slc17A7*[-/-] (VGLUT1 KO) (*Wojcik et al., 2004*) and *Slc17A7*[v/v] (VGLUT1[venus] knock-in) (*Herzog et al., 2011*) mice were maintained in C57BL/6N background and housed in 12/12 LD with ad libitum feeding. Every effort was made to minimize the number of animals used and their suffering. The experimental design and all procedures were in accordance with the European guide for the care and use of laboratory animals and approved by the ethics committee of Bordeaux Universities (CE50) under the APAFIS n°1692.

## Plasmids and viral vectors

From the Lentivector F(syn)W-RBN::VGLUT1[venus] previously published (*Siksou et al., 2013*), we engineered a series of point and deletion mutations of VGLUT1[venus] using conventional site directed mutagenesis protocols (see *Table 1*). Some experiments were performed using enhanced green fluorescent protein tagged synaptobrevin2 (F(syn)W-RBN::Syb2EGFP; *Siksou et al., 2013*). Lentiviral particles were generated by co-transfection of HEK-293T cells with the vector plasmid, a packaging plasmid (CMVD8.9 or CMV-8.74) and an envelope plasmid (CMV-VSVg) using Lipofectamine Plus (Invitrogen, Carlsbad, CA, USA) according to the manufacturer's instructions. 48 hr after transfection, viral particles were harvested in the supernatant, treated with DNaseI and $MgCl_2$, passed through a 0.45 μm filter and concentrated by ultracentrifugation ($\omega^2 t = 3,2\ 10^{10}\ rad^2/s$) and suspended in a small volume of PBS. Viral stocks were stored in 10 μl aliquots at −80° C before use. Viral titres were estimated by quantification of the p24 capsid protein using the HIV-1 p24 antigen immunoassay (ZeptoMetrix Corporation, Buffalo, NY, USA) according to the manufacturer's instructions. All productions were between 100 and 300 ng/μl of p24. Alternatively, recombinant adeno-associated virus vectors (AAV) were engineered with either WT or the transport deficient sVGLUT1 inserts (see

*Table 1*). We tagged these constructs with mCherry-miniSOG (*Qi et al., 2012*). Serotype 9 AAV particles were generated by transient transfection of HEK293T cells and viral stocks were tittered by QPCR on the recombinant genome as previously described (*Berger et al., 2015*). Both viral vectors control the expression of the inserts through the human synapsin promoter. For competition experiments with endophilinA1 and intersectin1 SH3 domains, we built fusions of the endophilinA1 SH3 domain (aa 290 to aa 352) or the SH3B domain of intersectin1 (aa 903 to 971) with the fluorescent protein mCerulean3 positioned at their N-terminus (*Markwardt et al., 2011*). All three SH3 domains were synthesized and the sequences checked (Eurofins genomics company). These fusions were cloned into the AAV shuttle plasmid.

## Hippocampal cell cultures and transgene expression

Hippocampal primary dissociated cultures were prepared from P0 mice. The hippocampi were dissected in ice-cold Leibovitz's L-15 medium (11415064; Gibco), and then incubated in 0.05% trypsin-EDTA (25300054, Gibco) for 15 min at 37°C. The tissues were washed with Dulbecco's Modified Eagle's Medium (DMEM, 61965026, Gibco) containing 10% FBS (CVFSVF0001, Eurobio), 1% Penicillin-streptomycin (15140122, Gibco). Cells were mechanically dissociated by pipetting up and down, and plated onto poly-L-lysine (P2636, Sigma) coated coverslips at a density of 20 000 cells/cm$^2$. Cells were grown in Neurobasal A medium (12349105, Gibco) containing 2% B27 supplement (17504044, Gibco), 0.5 mM Glutamax (35050038, Gibco), and 0.2% MycoZap plus-PR (VZA2021, Lonza) for 5 days in-vitro (DIV). From DIV 5–6, complete Neurobasal A medium was partially replaced to ½ by BrainPhys medium (*Bardy et al., 2015*), every two or three days. Imaging of live dissociated neuron cultures was performed at DIV 17–21 in culture medium with Hepes buffer (40 mM). Neurons were transduced at DIV1 or −2 with viral vectors diluted by a factor 1/1000. The viral expression levels were controlled by both western-blot (not shown), and fluorescence intensity to rescue VGLUT1 expression to endogenous VGLUT1 level. The bouton size and fluorescent intensity from the different VGLUT1 mutants were analyzed from the first frames the FRAP sequences. Images were processed by 'Find Edges', binary images were created through a threshold setting application to define the boundary of boutons. Area and background subtracted fluorescence intensity were measured. Integrated intentsity values were normalized to the value in WT boutons of the same group. The statistical significance of the bouton size and fluorescence intensity between the WT and VGLUT1 mutant rescues were evaluated with unpaired *t*-tests. For competition experiments with endophilinA1 and intersectin1 SH3 domains, plasmids were delivered by electroporation at DIV0 before plating (Nucleofector, Lonza).

## FRAP imaging

Fluorescence Recovery After Photobleaching (FRAP) experiments were performed to determine the mobility of Syb2 labeled SVs (for silent VGLUT1 mutant rescue experiments) or VGLUT1 labeled SVs (for VGLUT1 C-terminal mutants rescue experiments) at synapses. The mobile fraction of Syb2/VGLUT1 labeled SVs is the proportion of fluorescent material that can be replenished after photo bleaching. FRAP was performed using a spinning-disk confocal head Yokogawa CSU-X1 (Yokogawa Electric Corporation, Tokyo, Japan) mounted on an inverted Leica DMI 6000 microscope (Leica Microsystems, Wetzlar, Germany) and equipped with a sensitive EM-CCD QuantEM camera (Photometrics, Tucson, USA), and a FRAP scanner system (Roper Scientific, Evry, France). Surrounding the setup, a thermal incubator was set to 37°C (Life Imaging Services, Switzerland). Z-stacks of 4.8 $\mu m$ thickness were obtained with a piezo P721.LLQ (Physik Instrumente, Karlsruhe, Germany) at randomly selected fields from hippocampal cell culture with a 63×/1.4 numerical aperture oil-immersion objective. For each stack, five fluorescent boutons, distant from each other, were selected for bleaching. Three passes of the 491 nm laser (40 mW) for Syb2$^{EGFP}$ or two laser passes using the 491 nm laser (30 mW) and the 405 nm laser (10 mW) for VGLUT1$^{venus}$, were applied on the mid-plane of the stack and resulted in an average bleaching of 50% of the initial fluorescence intensity at boutons. The bleaching protocol for Venus/EYFP prevents the spontaneous recovery of fluorescence from a dark reversible photochemical state as previously reported (*Herzog et al., 2011*; *McAnaney et al., 2005*).

Fluorescence recovery was monitored every 30 s during the first 3 min and then every 5 min during the next 70 min. The entire FRAP procedure was controlled by MetaMorph (Molecular Devices,

Sunnyvale, USA). Image processing was automated using ImageJ macro commands (*Rasband, 1997*) (available at https://github.com/fabricecordelieres/IJ-Macro_FRAP-MM; copy archived at https://github.com/elifesciences-publications/IJ-Macro_FRAP-MM). Sum projections of the individual stacks, assembly and *x-y* realignment were applied, resulting in 32 bits/pixel sequences. Integrated fluorescence intensities of the five bleached boutons, and the cells in the field, as well as one background area were extracted. The background signal was subtracted, and data were normalized to the average baseline before bleaching (100%) and corrected for photobleaching against the cells. Experiments were discarded if photobleaching exceeded 60% (risk of phototoxicity). Fluorescence intensity of boutons was normalized to one before bleaching, and 0 right after bleaching. A double exponential function was used to fit the average of all normalized FRAP traces and the extra sum-of-squares F test was applied to compare the different best fits. Unpaired t-test was applied to analysis of bouton fluorescence intensity between different mutants and the corresponding WT data sets.

## Live cell imaging

Time-lapse experiments using the spinning disk confocal microscope were performed to quantify the SVs moving along the inter-synaptic axonal segments. Images were sampled at five frames/s for 30 s with 200 ms exposure time (151 frames in total). Synaptic boutons were saturated in order to allow better visualization of the dimmer fluorescent material moving along the axons. Quantification of the speed of moving clusters was performed with the KymoToolbox plugin in ImageJ (*Figure 4*; available at https://github.com/fabricecordelieres/IJ-Plugin_KymoToolBox) (*Zala et al., 2013*). In each sequence, eight axons segments of 10–15 µm were selected for particle tracking. Furthermore, the traffic at inter-synaptic segments was quantified by drawing line ROIs perpendicularly to the axon, and cumulating the integrated density values in each of the 151 images. Background was subtracted, and all values were divided by the average of 10 lowest values of the sequence to normalize for the differences of fluorescence intensity between different sets of experiments. All normalized values from each line selection were summed to evaluate the total amount of material going through the given cross-section. The statistical significance of the differences in cumulative traffic between the WT and VGLUT1 mutants (S540A and P554A) were evaluated with unpaired *t*-test.

## Electrophysiology

Dual whole-cell patch-clamp recordings were performed from DIV17 to DIV21. Patch pipettes (2–4 MΩ) were filled with the following intracellular solution (in mM): 125 CsMeSO$_3$, 2 MgCl$_2$, 1 CaCl$_2$, 4 Na2ATP, 10 EGTA, 10 HEPES, 0.4 NaGTP and 5 QX-314-Cl, pH was adjusted to 7.3 with CsOH. Extracellular solution was a standard ACSF containing the following components (in mM): 124 NaCl, 1.25 NaH$_2$PO$_4$, 1.3 MgCl$_2$, 2.7 KCL, 26 NaHCO$_3$, 2 CaCl$_2$, 18.6 Glucose and 2.25 Ascorbic acid. To record excitatory and inhibitory miniature currents (mEPSC and mIPSC), Tetrodotoxin (TTX) was added at 1 µM into an aliquot of the standard ACSF (Alomone labs). Cultures were perfused at 35°C with an ACSF perfusion speed of 0.02 mL/min and equilibrated with 95% O2/5% CO2. Signals were recorded at different membrane potentials under voltage clamp conditions for about 2 min (0 mV for inhibitory events and −70 mV for excitatory events) using a MultiClamp 700B amplifier (Molecular Devices, Foster City, CA) and Clampfit software. Recording at 0 mV in voltage clamp allowed us to confirm the effect of TTX on network activity. The recording of miniature events began 2 min after adding TTX and the extinction of synchronized IPSCs. Additional recordings were performed at membrane potentials of −20 mV, −40 mV, −60 mV and −80 mV. For drug treatment, CNQX was used at 50 µM and PTX at 100 µM and 2 min after drug addition, the condition was considered as stable.

## Electrophysiology analysis

Analyses were performed using Clampfit (Molecular Device), in which we created one mini-excitatory (−70 mV) and one mini-inhibitory (0 mV) template from a representative recording. Those templates were used for all recordings and analysis was done blind to the experimental group. We measured the number of mEPSCs at −70 mV and mIPSCs at 0 mV and their mean amplitude. Cell properties were monitored to get a homogenous set of cells, that is we analyzed the seal-test recordings of every cell (see *Supplementary file 2*) and calculated the capacitance from the Tau measured by Clampfit (*Supplementary file 2*). Cells with a leak current over −200 pA, and/or a membrane

resistance over −100MOhms were excluded from the analysis. Statistical analyses were performed using One-way ANOVA or Kruskal-Wallis test (* for p<0.05, ** for p<0.01 and *** for p<0.001).

## Antisera

The detection of wild type VGLUT1 was performed with rabbit polyclonal antiserum (*Herzog et al., 2011*; *Herzog et al., 2001*) whereas the detection of VGLUT1[venus] was done with a mouse monoclonal anti-GFP antibody (11814460001, Roche). Anti-VIAAT (131004, SYSY) and Anti-VGLUT2 (AB2251, Millipore) guinea pig polyclonal antisera were used. For electron microscopy we used a guinea pig anti-VGLUT1 polyclonal antiserum (AB5905, Merck) and a rabbit polyclonal antiserum against endophilinA1 (*Vinatier et al., 2006* ). Secondary HRP-coupled anti-rabbit, anti-mouse and anti-guinea pig antibodies were used for western blot detection (711-035-152, 715-035-150, 706-035-148, respectively, Jackson ImmunoResearch).

## Immunocytochemistry

Neuron cultures were washed with cold 1X PBS and fixed with 4% paraformaldehyde in 1X PBS for 5 min at room temperature. Immunostainings were performed as previously described (*Herzog et al., 2001*). Wide-field pictures were acquired using an epifluorescence Nikon Eclipse NIS-element microscope with the 40x objective. The ImageJ software was used to quantify the fluorescence intensity. A mean filter, background subtraction, and threshold were applied to cover the punctate signals and generate a selection mask. The mean density values were extracted. The averages of 5 frames per culture were probed and more than three cultures per age were measured.

## Immunohistochemistry and electron microscopy

VGLUT1 and EndophilinA1 were simultaneously detected by combination of immunoperoxidase and immunogold methods, respectively, on brain sections at the ultrastructural level.

## Animals and tissue preparation

The mice were deeply anesthetized with sodium chloral hydrate and then perfused transcardially with 0.9% NaCl, followed by fixative consisting of 2% paraformaldehyde (PFA) with 0.2% glutaraldehyde in 0.1M phosphate buffer (PB), pH 7.4, at 4°C. The brain was quickly removed and left overnight in 2% PFA at 4°C. Sections from cerebellum and caudate putamen were cut on a vibrating microtome at 70 mm and collected in PBS (0.01M phosphate, pH 7.4). To enhance the penetration of the immunoreagents in the preembedding procedures, the sections were equilibrated in a cryoprotectant solution (0.05MPB, pH 7.4, containing 25% sucrose and 10% glycerol) and freeze-thawed by freezing in isopentane cooled in liquid nitrogen andthawed in PBS. The sections were then preincubated in 4% normal goat serum (NGS) in PBS.

## Double detection of VGLUT1 and EndophilinA1 by combined preembedding immunoperoxidase and immunogold methods

The brain sections were incubated in 4% NGS for 30 min and then in a mixture of VGLUT1 (1:5000) and EndophilinA1 (1:500) antibodies, supplemented with 1% NGS overnight at RT. The sections were then incubated with goat anti-guinea-pig IgGs conjugated to biotin (1:200), washed in PBS and incubated in a mixture of avidin–biotin–peroxidase complex (ABC), (1:100; Vector Laboratories, Burlingame, CA) and goat anti-rabbit IgGs conjugated to gold particles (1.4 nm diameter; Nanoprobes, Stony Brook, N Y; 1:100 in PBS/BSA C) for 2 hr in PBS/BSA C. The sections were then washed in PBS and post-fixed in 1% glutaraldehyde in PBS. After washing (PBS; sodium acetate buffer, 0.1M, pH 7.0), the diameter of the gold immunoparticles was increased using a silver enhancement kit (HQ silver; Nanoprobes) for 5 min at RT in the dark. After washing (2xPBS, 1xTB 0.05M, pH 7.6), the immunoreactive sites VGLUT1 were revealed using DAB. The sections were then stored in PB and processed for electronmicroscopy.

## Preparation for electron microscopy

The sections were post-fixed in osmium tetroxide (1% in PB, 0.1M,pH7.4) for 10 min at RT. After washing, they were dehydrated in an ascending series of dilutions of ethanol that included 1% uranyl acetate in 70% ethanol. They were then treated with propylene oxide and equilibrated in resin

overnight (Durcupan AC M; Fluka, Buchs, Switzerland), mounted on glass slides, and cured at 60°C for 48 hr. The immunoreactive areas identified on thick sections were cut in semithin sections (1-mm-thick), then in ultrathin sections on a Reichert Ultracut S. Ultrathinsections were collected on pioloform-coated single slot copper grids. The sections were stained with lead citrate and examined in a Philips C M10 electron microscope.

## Biochemistry

All steps were performed at 4°C or on ice. Brains of wild type adult mice were dissected for the collection of brain regions. The samples were treated with homogenization buffer (0.32 M sucrose, 4 mM HEPES pH 7.4). Cultures expressing the different VGLUT1 mutants were collected on DIV 17 with $1 \times$ PBS. Both buffers were supplemented with protease inhibitor cocktail (539134, Millipore) and Halt phosphatase Inhibitor Cocktail (78420, Thermo Fisher Scientific). When necessary, protein samples were treated with alkaline phosphatase prior to the biochemical analysis. FastAP Thermo-sensitive Alkaline Phosphatase (one unit/μl) and FastAP $10 \times$ buffer (EF0654, Thermo Scientific) were added to samples and incubated for 1 hr at 37°C. Sodium dodecylsulfate polyacrylamide gel electrophoresis (SDS-PAGE) and phosphate affinity SDS-PAGE (*Kinoshita et al., 2006*) were conducted according to standard methods. For phosphate affinity SDS-PAGE ($Mn^{2+}$-Phos-tag SDS-PAGE), 25 μM Phostag (AAL-107, Wako) and 0.1 mM $MnCl_2$ were added to the resolving gel before polymerization. Western blotting was performed according to standard procedures using HRP-coupled secondary antibodies for qualitative detection. Chemi-luminescence signals were visualized with ChemiDoc MP System (Bio-Rad) using SuperSignal West Dura Extended Duration Substrate (34075, Thermo Scientific).

## Acknowledgements

Elisa Luquet, Sally Wenger, and Charlène Josephine for excellent technical support. Nils Brose for a strong support throughout the project. Serge Marty and Martin Oheim for fruitful discussions. Christian Rosenmund, Roger Tsien, Mark A Rizzo, for sharing reagents. Several experiments required the use of Bordeaux University/CNRS/INSERM core facilities: Bordeaux Imaging Center (member of France BioImaging supported by the French National Research Agency; ANR-10-INBS-04); Vect'UB viral vector facility; Biochemistry and Biophysics of Proteins core facility; Mouse breeding facility. XMZ was supported by the Erasmus Mundus ENC program and the labex BRAIN extension grant (ANR-10-LABX-43 BRAIN). Funding from the Agence Nationale de la Recherche (ANR-12-JSV4-0005-01 VGLUT-IQ ; ANR-10-LABX-43 BRAIN; ANR-10-IDEX-03–02 PEPS SV-PIT to EH and PSY-VGLUT 09-MNPS-033 to SEM). KS and MFA were supported by the Fondation pour la Recherche Médicale (FDT20120925288 and ING20150532192 respectively).

## Additional information

### Funding

| Funder | Grant reference number | Author |
|---|---|---|
| Agence Nationale de la Recherche | ANR-12-JSV4-0005-01 VGLUT-IQ | Etienne Herzog |
| Agence Nationale de la Recherche | ANR-10-LABX-43 BRAIN | Yann Humeau Etienne Herzog |
| Agence Nationale de la Recherche | ANR-10-IDEX-03-02 PEPS SV-PIT | Etienne Herzog |
| Agence Nationale de la Recherche | PSYVGLUT 09-MNPS-033 | Salah El Mestikawy |
| Agence Nationale de la Recherche | ANR-10-INBS-04 | Fabrice P Cordelières |
| European Commission | Erasmus-Mundus European Neuroscience Campus Program | Xiao Min Zhang Etienne Herzog |

| Fondation pour la Recherche Médicale | FDT20120925288 | Kätlin Silm |
| Fondation pour la Recherche Médicale | ING20150532192 | Maria Florencia Angelo Etienne Herzog |

The funders had no role in study design, data collection and interpretation, or the decision to submit the work for publication.

## Author contributions
Xiao Min Zhang, Formal analysis, Investigation, Methodology, Writing—original draft; Urielle François, Formal analysis, Investigation, Methodology, Writing—original draft, Writing—review and editing; Kätlin Silm, Formal analysis, Investigation, Methodology, Writing—review and editing; Maria Florencia Angelo, Maria Victoria Fernandez-Busch, Resources, Formal analysis, Supervision, Investigation, Methodology, Writing—review and editing; Mona Maged, Formal analysis, Investigation; Christelle Martin, Resources, Supervision, Methodology, Project administration; Véronique Bernard, Investigation, Methodology, Writing—review and editing; Fabrice P Cordelières, Software; Melissa Deshors, Resources, Project administration; Stéphanie Pons, Alexis Pierre Bemelmans, Resources, Methodology, Writing—review and editing; Uwe Maskos, Resources, Supervision, Funding acquisition; Sonja M Wojcik, Resources, Writing—review and editing; Salah El Mestikawy, Supervision, Funding acquisition, Project administration, Writing—review and editing; Yann Humeau, Conceptualization, Formal analysis, Supervision, Funding acquisition, Project administration, Writing—review and editing; Etienne Herzog, Conceptualization, Formal analysis, Supervision, Funding acquisition, Writing—original draft, Project administration

## Author ORCIDs
Fabrice P Cordelières (iD) http://orcid.org/0000-0002-5383-5816
Etienne Herzog (iD) https://orcid.org/0000-0002-0058-6959

## Ethics
Animal experimentation: The experimental design and all procedures were in accordance with the European guide for the care and use of laboratory animals, approved by the ethics committee of Bordeaux Universities (CE50), and registered with the French Ministry for Research under the APAFIS n°1692. Every effort was made to minimize the number of animals used and their suffering.

## Decision letter and Author response
Decision letter https://doi.org/10.7554/eLife.50401.036
Author response https://doi.org/10.7554/eLife.50401.037

# Additional files

## Supplementary files
• Supplementary file 1. Supplementary tables collating statistical analysis.
DOI: https://doi.org/10.7554/eLife.50401.032

• Supplementary file 2. Seal test recording of every cell in the electrophysiology analysis.
DOI: https://doi.org/10.7554/eLife.50401.033

• Transparent reporting form DOI: https://doi.org/10.7554/eLife.50401.034

## Data availability
Raw measures and intermediate data processing of images and electrophysiology traces are submitted in source files appended to this submission. Source images and electrophysiology traces reported in this study are fully available upon request to the corresponding author (Etienne Herzog—https://orcid.org/0000-0002-0058-6959).

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
