## [Decision Letter]

**Acceptance summary:**

It was shown previously that the vesicular glutamate transporter VGLUT1 is involved in clustering of synaptic vesicles in the synapse, and that its loss increases the exchange of vesicle between the synapse and extrasynaptic compartments. Here, the authors show in an elegant series of experiments that the C-terminally located proline-rich motif of VGLUT1 is responsible for these effects. In particular, the exchange of SVs between the synapses depends on the second proline-rich motif. Moreover, the authors show that the effects are mediated by the recruitment of Intersectin1 to VGLUT1/endophilinA1 complexes, thus establishing a role of SH3/proline-rich interactions in the dynamics of SVs. These effects are specific for VGLUT1 (VGLUT2 lacks such a proline rich domain) and are independent of the transport activity of VGLUT1. Taken together, these data establish an important function of VGLUT in balancing vesicle clustering with vesicle exchange between pools.

**Decision letter after peer review:**

Thank you for submitting your article "A poly-proline motif on VGLUT1 reduces synaptic vesicle super-pool and spontaneous release frequency" for consideration by *eLife*. Your article has been reviewed by three peer reviewers, one of whom is a member of our Board of Reviewing Editors, and the evaluation has been overseen by Gary Westbrook as the Senior Editor. The following individuals involved in review of your submission have agreed to reveal their identity: Dragomir Milovanovic (Reviewer #2).

The reviewers have discussed the reviews with one another and the Reviewing Editor has drafted this decision to help you prepare a revised submission.

Summary:

This study is founded on the previous work by some of the authors (Siksou et al., 2013) in which evidence is provided that deletion of the vesicular glutamate transporter VGLUT1 leads to a decrease in synaptically clustered synaptic vesicles and a concomitant increase in the so-called "superpool" of synaptic vesicles, defined by a rapid exchange of SVs between synapses and extrasynaptic compartments. Zhang and colleagues focus on dissecting the molecular mechanisms responsible for these effects while taking advantage of previously generated VGLUT1 knock-out and VGLUT1-Venus knock-in mouse lines. Using a combination of imaging and electrophysiology, the authors show in an elegant series of experiments that the C-terminally located proline-rich motif is responsible for the effects. In particular, the exchange of SVs between the synapses, part of a so-called SV super-pool, depends on the second proline-rich motif (i.e., PP2). Moreover, the authors show that the effects are mediated by the recruitment of Intersectin1 to VGLUT1/endophilinA1 complexes, thus establishing a role of SH3/proline-rich interactions in the dynamics of SVs. These effects are specific for VGLUT1 (VGLUT2 lacks such a proline rich domain) and are independent of the transport activity of VGLUT1.

All reviewers agree that the work is interesting, that the study is well-controlled and that the main conclusions are well supported by the data. Moreover, the reviewers agree that no additional experiments are essential for acceptance.

Reviewer #1:

The interpretation of the experiments is mainly dependent on the interpretation of the FRAP experiments. Here, I am a bit confused by the description of the authors because they state that the rate of recovery is altered. However, to me it appears that the rate is unchanged but rather the extent of the recovery is changed, which would be in agreement with the interpretation that in the absence of the polyproline motif there are less clustered and thus immobile (and non-exchangeable) SVs. This correlates with mepp frequency – less vesicles in the cluster leads to lower mepp frequency. Second, it would strengthen the interpretation of the data if rescue experiments were shown with VGLUT2, which does not contain the PP stretch at the C-terminus. Along similar lines, it would be interesting to test whether mutation of the dileucine motif important for endocytosis has an effect on the phenotype studied here. Finally, for the tripartite complex proposed here (VGLUT1, endophilinA1, intersectin) some biochemical support (e.g. by co-IPs or similar) should be provided.

Finally, while it is certainly somewhat subjective, this reviewer finds the manuscript difficult to read. This is partially due to the integration of all statistical data into the main text. Perhaps there is a better way to present these data (e.g. in supplemental tables) to improve readability. Similarly, the first part of the discussion is essentially a repetition of the results – this can be considerably shortened.

Reviewer #2:

1) It remains unclear whether the same mechanism-multivalent, low-affinity interactions-underlies both the release probability of SVs and the exchange of vesicles between the clusters. Would a chimeric VGLUT1 construct that contains a string of four (or six) PP2 motifs further both reduce the mEPSC frequency and decrease the rate of SV exchange between the pools? What is happening with mEPSC frequency in the competition assays (Figure 5)?

2) In a simplified model, one could hypothesize that the exchange of SVs contains, at least, two distinct steps: (i) the 'escape' of an SV from the cluster, and (ii) the transport of an SV between the synapses. (Further steps are very probable, but would go beyond the main focus.) What step would the interactions described in the manuscript affect? The velocity of SV trafficking along the axon appears unchanged (Figure 4C-E). Are there changes in the mobility of SVs within a cluster (e.g., fluorescence recovery after bleaching a small region of bouton)? An interplay between actin/myosin V was shown to alter the SV exchange (Gramlich and Klyachko 2017, PMID: 28249156). Does sequestering of soluble actin or stabilization of actin polymers affect the rate of SV exchange in VGLUT knock-out cells and VGLUT-PP2 mutants?

3) A term "proline-rich motif" is more commonly used than "poly-proline motif". Authors may consider changing this.

Reviewer #3:

1) The paper is very difficult to read. The statistical description should be restricted to the figure legends and not given in the main text.

2) The interaction between endophilin /VGLUT1 and intersectin1 SH3B domain has been described elsewhere and mainly mediates endocytosis of VGLUT1. The SV super-pool appears to contain mobile SVs. So the here described function would involve that binding of endophilin to VGLUT1 and intersectin1 to this complex are not restricted to a plasma membrane association. However, to my knowledge there are no data provided if either endophilin or intersectin1 are (transiently) associated with SVs. Is there any evidence at the EM level?

3) If endophilin/VG1 interaction is crucial for the exchange with the SV super-pool, it should be impaired by deletion of endophilin (si RNA interference of endophilin KO mice).

4) Intersectin1 has five SH3 domains some of them appear to interact with VGLUT1 in addition (besides the indirect intersectin1-SH3B/endophilin interaction). Are there any evidence on the contribution of whole intersecin1 and/or the other SH3 domains?

5) Is the exchange between the SV super-pool and SV clusters in axon terminals restricted to VGLUT1 containing SVs? This should be at least discussed.

6) Figure 5 C is too complex. It would benefit from an additional block diagram focusing on one time point.

---

## [Author Response]

Reviewer #1:The interpretation of the experiments is mainly dependent on the interpretation of the FRAP experiments. Here, I am a bit confused by the description of the authors because they state that the rate of recovery is altered. However, to me it appears that the rate is unchanged but rather the extent of the recovery is changed, which would be in agreement with the interpretation that in the absence of the polyproline motif there are less clustered and thus immobile (and non-exchangeable) SVs. This correlates with mepp frequency – less vesicles in the cluster leads to lower mepp frequency.

Indeed, this point is very important and deserves more precision in the final version. 2 types of FRAP experiments are performed throughout our study. Syb2^EGFP^ FRAP in Figure 1A-C and Figure 3. The advantage of using Syb2^EGFP^ is that we probe SV exchange with a different protein than VGLUT1. Key for the WT vs KO comparison and to avoid possible conundrum due to the fact that we probe SVs with the same protein that is our subject of study. The disadvantage is that Syb2 has a significant plasma membrane pool, possibly slightly different intracellular trafficking compared to VGLUT1 and finally is over-expressed. In experiments of Figure 1DE and 4, we FRAP VGLUT1^venus^ in rescue and over-expression experiments. The advantage of the rescue experiment is that we replace VGLUT1 at near endogenous levels and the only alterations between conditions may be brought by the mutations tested. Both types of experiment produce consistent qualitative results and trends. The reviewer is right that the changes observed in Syb2^EGFP^ are restricted to shifts in the plateau levels. The fast and slow half-lives of recovery being equivalent between respective conditions of the 2 figures (1C and 3B). However, using VGLUT1^venus^ rescue constructs we see that the slow half life of FRAP recovery is massively shortened by the P554A mutant compared to WT (Figure 4). Of note, the WT kinetic is similar to that reported before from VGLUT1^venus^ mice (Herzog et al., 2011). We now report in detail the kinetic parameters for the experiment of Figure 4 as we believe they are strongly informative of the molecular mechanism of PP2, and devoid of major artifact. As a consequence, we believe that PP2 of VGLUT1 participate to interactions that change the rate of exchange of SV between the cluster and the axonal super-pool, as well as changes the level of clustered vs super-pool sizes. The text has been modified to strengthen this point.

Finally, we wish to make clear that mEPSC frequency increases when clustered SVs are less abundant and mobile ones more abundant. PRD2 of VGLUT1 acts as a negative regulator of mEPSC frequency.

Second, it would strengthen the interpretation of the data if rescue experiments were shown with VGLUT2, which does not contain the PP stretch at the C-terminus.

Unfortunately, at the time we did the FRAP experiments comparing VGLUT1 and -2, we had not established the electrophysiology part. When performing the electrophysiology part we focused our attention on having the right controls for the sVGLUT1 mutant and the proline point mutation. The reviewer is right that this experiment would be a nice complement, yet in the present state of our laboratory, it would require a 6 month delay (Indeed, the entire mouse colonies have been frozen prior to moving into our new facility of Bordeaux Neurocampus. The VGLUT1 KO mouse line has not yet been revived …). Hence we would prefer not to add this experiment.

Along similar lines, it would be interesting to test whether mutation of the dileucine motif important for endocytosis has an effect on the phenotype studied here.

We have not included such mutants because they affect drastically the integration of VGLUTs into SV membranes and induce an abnormally large plasma membrane pool of VGLUTs (See Voglmaier et al., 2006; and Foss et al., 2013). In VGLUT1^venus^ FRAP we expect a strong effect on our phenotype due to the wrong location of the transporters but not to a genuine change in the behavior of SV pools. In Syb2^EGFP^ FRAP we expect a phenotype close to the one of the VGLUT1 KO as a significant fraction of the VGLUT1 protein will be absent from SV membranes.

Finally, for the tripartite complex proposed here (VGLUT1, endophilinA1, intersectin) some biochemical support (e.g. by co-IPs or similar) should be provided.

The SH3-SH3 interaction of endophilinA1 and intersectin 1 has been elegantly shown and documented in Pechstein et al., 2015. In their experiments they show that only through this SH3-SH3 interaction, it is possible to assemble a tripartite complex with endophilinA1, intersectin1 and VGLUT1 (Figure 4 from Pechstein et al., 2015). The possibility of this interaction was confirmed by investigations from the team of Ahnert-Hilger in 2018 (Richter et al., 2018). Yet we think that this tripartite complex may be transient and relatively weak. We tried to tackle this issue with fluorescence lifetime imaging (FLIM) between venus and cerulean molecules. We obtained interesting FLIM trends from our test constructs, but unfortunately, technical issues with control assays prevent us from displaying the corresponding results at this stage.

Finally, while it is certainly somewhat subjective, this reviewer finds the manuscript difficult to read. This is partially due to the integration of all statistical data into the main text. Perhaps there is a better way to present these data (e.g. in supplemental tables) to improve readability. Similarly, the first part of the discussion is essentially a repetition of the results – this can be considerably shortened.

The point is well taken. We edited our Results section to smoothen reading, all statistics were added to a supplemental table. We also rewrote the first part of the discussion to shorten it and focus on discussing the current knowledge on VGLUT1 structure/function relationships.

Reviewer #2:1) It remains unclear whether the same mechanism-multivalent, low-affinity interactions-underlies both the release probability of SVs and the exchange of vesicles between the clusters. Would a chimeric VGLUT1 construct that contains a string of four (or six) PP2 motifs further both reduce the mEPSC frequency and decrease the rate of SV exchange between the pools?

Unfortunately we didn’t push our study in that direction as we rather focused on trying to find the smallest perturbation required to mimic the phenotype of VGLUT1 knockout neurons. Yet our experiment with VGLUT1 overexpression suggests that the effect of PP2 is dose dependent on SV exchange. As discussed in subsection “Mammalian VGLUT1 acts as a dual regulator of glutamate release”, our data are in line with those of Weston (Weston et al., 2011) and suggest that endophilin promotes release probability by itself, while VGLUT1 scavenges endophilin and reduces both SV mobility and inhibits release.

What is happening with mEPSC frequency in the competition assays (Figure 5)?

This is an extremely interesting question however we could not address this question with the methodology used.

Indeed, we induced a strong overexpression of the SH3 constructs using conventional transfection methods. This ensured the right conditions to have a strong competition of our tools with endogenous complexes. However only a few neurons per plate were efficiently transfected. To measure mEPSC changes we need to have a majority of cells exhibiting the same “genotype” in the culture so that the phenotype measured on a given patched cell is relevant.

2) In a simplified model, one could hypothesize that the exchange of SVs contains, at least, two distinct steps: (i) the 'escape' of an SV from the cluster, and (ii) the transport of an SV between the synapses. (Further steps are very probable, but would go beyond the main focus.) What step would the interactions described in the manuscript affect? The velocity of SV trafficking along the axon appears unchanged (Figure 4C-E). Are there changes in the mobility of SVs within a cluster (e.g., fluorescence recovery after bleaching a small region of bouton)? An interplay between actin/myosin V was shown to alter the SV exchange (Gramlich and Klyachko 2017, PMID: 28249156). Does sequestering of soluble actin or stabilization of actin polymers affect the rate of SV exchange in VGLUT knock-out cells and VGLUT-PP2 mutants?

Actin/myosin cytoskeleton is certainly important for axonal transport of SVs in the super-pool and possibly for the organization of subcellular compartments of the terminal and this should be true also for VGLUT mutants yet we don’t have any evidence for a specific interaction of VGLUT1 with the actin cytoskeleton.

According to Shupliakov and collaborators, Intersectin interacts with synapsins to promote SV reclustering after endocytosis (Winther et al., 2015). Our interpretation of the current data is similar and we propose that VGLUT1 PP2 promotes this pathway rather than an escape of SVs to the axonal super-pool. It remains to understand whether SV mobility inside the vesicular cluster is also influenced by the PP2 interactions. This could be addressed in a follow up study using sptPALM approaches rather than FRAP. Indeed to our knowledge, FRAP resolution is not high enough to address questions at the sub-micron scale. We adapted these points in the discussion of the paper.

3) A term "proline-rich motif" is more commonly used than "poly-proline motif". Authors may consider changing this.

This was systematically corrected in the text. The abbreviations PP1 / PP2 were changed to PRD1 / PRD2 in the text but not in this correspondence.

Reviewer #3:1) The paper is very difficult to read. The statistical description should be restricted to the figure legends and not given in the main text.

The point is well taken. Statistical description has been moved to a supplemental table and figure legends.

2) The interaction between endophilin /VGLUT1 and intersectin1 SH3B domain has been described elsewhere and mainly mediates endocytosis of VGLUT1. The SV super-pool appears to contain mobile SVs. So the here described function would involve that binding of endophilin to VGLUT1 and intersectin1 to this complex are not restricted to a plasma membrane association. However, to my knowledge there are no data provided if either endophilin or intersectin1 are (transiently) associated with SVs. Is there any evidence at the EM level?

We thank the reviewer for this interesting comment. At first we wish to clear one point. Endocytosis of VGLUT1 is mainly driven by several dileucine motifs as elegantly shown by the groups of Edwards and Voglmaier (Voglmaier et al., 2006, Foss et al., 2013). The PP2 motif was only pointed as a positive regulator of SV endocytosis during prolonged stimulations and as a negative regulator of SV release probability. Literature on Intersectin points to a role in clathrin uncoating and SV clustering after endocytosis. As pointed by reviewer 2, one may consider a step of sorting for endocytosed SVs between escaping to the super-pool or integrating the cluster for another maturation to local exocytosis. Our data suggest that the gain of VGLUT1 PP2 favors SV clustering and reduces SV escape to the super-pool by increasing the recruitment of intersectin1 to SVs. To our knowledge, there is no report of immunostaining electron micrographs for EndoA1 in the literature. We previously published immunofluorescence stainings of EndoA1 with a custom made antibody (Vinatier et al., 2006). We have some electron micrographs from that time that clearly show the localization of EndoA1 over SV clusters in cerebellar mossy fiber synapses and cortico-striatal synapses. We have now included this figure in the supplements (Figure 5—figure supplement 1, and corresponding methods). Furthermore, intersectin was shown over SV clusters in lamprey (Figure 3A in Evergren et al., 2007). Hence we think that our data supports that SV clusters are maintained at least in part through a network of weak PRD/SH3 interactions. This is strengthen by the occurrence of PRD2 on VGLUT1 through an interaction with both endophilin and intersectin. This discussion has been adapted to the present version of the text. Specifically, reference to the ultrastructural localization of endophilinA1, intersectin and amphiphysin was added.

3) If endophilin/VG1 interaction is crucial for the exchange with the SV super-pool, it should be impaired by deletion of endophilin (si RNA interference of endophilin KO mice).

In fact, Endophilin/VG1 interaction is not crucial for the exchange of SVs with the super-pool, It is rather a negative regulator of SV exchange between the synaptic clusters and the super-pool. The issue with endophilinA1 knock out is that much compensation is provided by the other isoforms of endophilin. Hence, available data come from the triple endophilin KO mice with a massive phenotype blocking clathrin uncoating (Milosevic et al., 2011). Such reverse experiment to ours using endophilinA1 alteration is therefore not trivial to implement.

4) Intersectin1 has five SH3 domains some of them appear to interact with VGLUT1 in addition (besides the indirect intersectin1SH3B/endophilin interaction). Are there any evidence on the contribution of whole intersecin1 and/or the other SH3 domains?

We are grateful to the reviewer for pointing this to our attention. It is right that Richter and colleagues did show a convincing experimental set supporting the interaction of VGLUT1 with ITSN1-SH3A and -SH3B (Richter et al., 2018). The interaction with SH3B could be indirect through EndophilinA1-SH3 as elegantly shown by Pechstein et al., (EMBO rep, 2015). We overlooked this point in the previous version of the discussion and corrected this in the current version. Our data in Figure 5 suggest that ITSN1-SH3B is sufficient to displace ITSN1 and phenocopy the VGLUT1 KO phenotype. Hence we do believe that ITSN1 full sequence is required for the reduction of SV exchange between the cluster and the super-pool. However we have no evidence that ITSN1-SH3A is involved in the process. Indeed many aspects remain obscure at this stage. For instance, we deployed many efforts to unravel functions of other parts of VGLUT1 c-terminus such as PP1 with no success yet. We probably need other assays than the FRAP to access these phenotypes/functions.

5) Is the exchange between the SV super-pool and SV clusters in axon terminals restricted to VGLUT1 containing SVs? This should be at least discussed.

Clearly not in our opinion even though a formal exploration of SV super-pool in other neuronal types is missing to our knowledge. Yet the work of Wierenga and colleagues strongly suggests the existence of SV exchange at GABAergic axons (Wierenga et al., 2008). Also recent work shows that SV mobility in the axon is regulated by GPCR neuromodulation (Patzke et al., 2019). Once again our finding is that VGLUT1 PP2 acts as a negative regulator of these exchanges. We now state this in the Introduction.

6) Figure 5 C is too complex. It would benefit from an additional block diagram focusing on one time point.

The point is well taken. We have replaced the FRAP curves by a block diagram focusing on the end point of the FRAP experiment and rewritten the figure legend accordingly.